# $k$NN Prompting: Beyond-Context Learning with Calibration-Free Nearest Neighbor Inference

**Benfeng Xu**[1], **Quan Wang**[2], **Zhendong Mao**[14]*, **Yajuan Lyu**[3], **Qiaoqiao She**[3], **Yongdong Zhang**[14]

[1]University of Science and Technology of China, Hefei, China
[2]Beijing University of Posts and Telecommunications, Beijing, China   [3]Baidu Inc., Beijing, China
[4]Institute of Artificial Intelligence, Hefei Comprehensive National Science Center, China
benfeng@mail.ustc.edu.cn, zdmao@ustc.edu.cn

## Abstract

In-Context Learning (ICL), which formulates target tasks as prompt completion conditioned on in-context demonstrations, has become the prevailing utilization of LLMs. In this paper, we first disclose an actual predicament for this typical usage that it can not scale up with training data due to context length restriction. Besides, existing works have shown that ICL also suffers from various biases and requires delicate calibration treatment. To address both challenges, we advocate a simple and effective solution, $k$NN Prompting, which first queries LLM with training data for distributed representations, then predicts test instances by simply referring to nearest neighbors. We conduct comprehensive experiments to demonstrate its two-fold superiority: 1) **Calibration-Free**: $k$NN Prompting does not directly align LLM output distribution with task-specific label space, instead leverages such distribution to align test and training instances. It significantly outperforms state-of-the-art calibration-based methods under comparable few-shot scenario. 2) **Beyond-Context**: $k$NN Prompting can further scale up effectively with as many training data as are available, continually bringing substantial improvements. The scaling trend holds across 10 orders of magnitude ranging from 2 shots to 1024 shots as well as different LLMs scales ranging from 0.8B to 30B. It successfully bridges data scaling into model scaling, and brings new potentials for the gradient-free paradigm of LLM deployment. Code is publicly available[1].

## 1 Introduction

Large language models (LLMs), when scale up to billions of parameters, have demonstrated remarkable capabilities in a wide range of NLP tasks (Radford et al., 2019; Brown et al., 2020). However, such models are prohibitively expensive to train with most of the research- or consumer-level devices, though some of them are already publicly available (Zhang et al., 2022). As a result, it is now an emerging paradigm that LLMs are hosted in a remote data center while accessed by end users or applications via simple API requests[2]. The typical usage of LLM under such paradigm is In-Context Learning, where LLM reads and completes a prompt sequence as how it is pretrained on massive text corpora. The

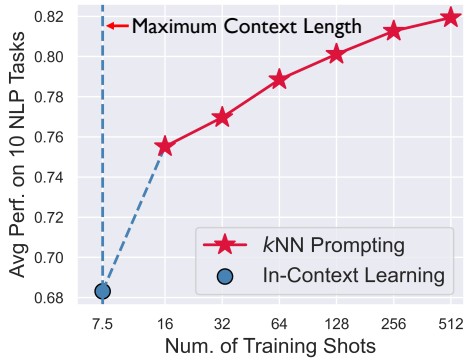

Figure 1: $k$NN Prompting brings substantial improvements over standard ICL, and can continually scale up beyond the context with as many data as are available. Conducted with GPT XL.

---

*Corresponding author.

[1]https://github.com/BenfengXu/KNNPrompting
[2]https://openai.com/api/, https://gpt3demo.com/

prompt is constructed by concatenation of several training examples and a test instance, and the prediction is obtained by mapping the LLM word continuations back to label space.

It is widely investigated and acknowledged that modern neural networks generally perform better w.r.t. increased training data. Specifically, there exists a power law between expected model performance and available data scale (Hestness et al., 2017; Rosenfeld et al., 2020). For ICL, it is also empirically observed that the performance continually improves when more training examples are prepended into the prompt (Brown et al., 2020). However, such improvements are quickly prevented by the predicament of context length restriction, as language models are designed and trained to only process sequences within a fixed length, which is in fact 1024 or 2048 tokens. In order to utilize more training data, several works try to select the most relevant examples to compose the prompt before querying LLM (Liu et al., 2022b; Rubin et al., 2022), but still only in-context examples can actually participate the LLM inference while most training data are discarded beforehand, thus providing marginal data scaling benefits. Besides, their reliance on external retriever also incurs further complications. As a consequence, such a situation poses a serious challenge for many practical scenarios where more than a few training data are available.

Another vulnerability of ICL is the severe bias existed in the output distribution of LLMs, which results in considerable performance degradation (Holtzman et al., 2021) and instability (Lu et al., 2022) as shown in existing works. Accordingly, many have proposed various ways to calibrate the output distribution (Zhao et al., 2021; Jiang et al., 2021; Min et al., 2022a). For example, Zhao et al. (2021) measure such bias by probing LLM with a "*NA*" example and record the according prior. However, as LLMs are pretrained on general-domain natural language, its capability to complete a fabricated prompt is essentially not aligned with downstream task-specific label space. As a consequence, such calibration-based methods can only alleviate the bias to a limited extent.

In this paper, we advocate a simple and effective solution, $k$NN Prompting, to address both challenges. Specifically, we assign training data into a demonstration set and an anchor set. We append each anchor example into the prompt and query LLM, then instead of aligning word continuations with labels, we collect the language modeling probability as distributed representation and cache it into a local datastore. At inference time, for each test instance, we similarly obtain its representation and match it against the maintained datastore to make predictions. In general, the proposed framework enables both calibration-free optimization because it avoids forced input-label alignment, and beyond-context learning because the anchor set allows utilization of unlimited training data.

We conduct comprehensive experiments using 10 established text classification tasks to demonstrate the significant superiority of $k$NN Prompting across various scenarios and against competitive opponents: 1) Under few shot scenario where training data is very limited and fits in the context, $k$NN Prompting outperforms state-of-the-art calibration-based methods by considerable margin (up to +7.07). 2) Under low resource or fully supervised scenario where training data can not fit in the context, $k$NN Prompting further exhibits its major advantage. It can effectively scale up with as many training data as are available across 10 orders of magnitude (2 shots~1024 shots, see Figure 1 for illustration) as well as different LLMs scales (0.8B~30B). Specifically, with only 32 shots training data, it dramatically improves ICL by +13.58 in average score at its most, and achieves absolute improvements up to +18.84 under fully supervised setting. We also provide formal explanation on the intrinsic mechanism of effectiveness, as well as detailed analyses regarding its robustness and choices of design. Accompanied with these appealing aspects, $k$NN Prompting is in general a promising solution that bridges the benefits of data scaling into model scaling to take the gradient-free paradigm of LLM deployment one step further.

## 2 BACKGROUND: IN-CONTEXT LEARNING

In this section, we formulate the task and recap the ICL baseline. Assuming a target task with training data set $\mathcal{T} = \{(x_i, y_i)\}$, and $Y$ as its categorical label space. At inference time, the model is asked to predict $y_{\text{test}}$ given test instance $x_{\text{test}}$. We then denote an LLM $\boldsymbol{\theta}$ that is pretrained with a standard language modeling objective. At employment, it samely predicts a probability distribution $p(w_t | \boldsymbol{w}_{<t}, \boldsymbol{\theta})$ for the next token at $t$-th position conditioned on previous context $\boldsymbol{w}_{<t}$.

In-context learning first formulates training examples $\{(x_i, y_i)\}$ in the format of input-label mapping via intuitive templates (see Appendix F for illustration), and concatenates them into a natural

| | SST2 | SUBJ | MPQA | AGNews | CB | CR | DBPedia | MR | RTE | TREC |
|---|---|---|---|---|---|---|---|---|---|---|
| **Num. of Shots (TP)** | 20 (2%) | 12 (1%) | 39 (0%) | 3 (0%) | 2 (0%) | 14 (4%) | 1 (77%) | 14 (4%) | 4 (0%) | 8 (1%) |
| $M_T$ | 16 | 8 | 32 | 2 | 2 | 8 | 1 | 8 | 4 | 8 |

Table 1: Maximum number of training shots (per class) allowed by 1024 tokens of context. Calculated using GPT2 tokenizer. Inside the parentheses are Truncation Probability (TP, i.e., whether or not truncated, restricted below 5%). We set $M_T$ for each task $T$ in our experiments for simplicity.

language sequence along with the test instance to construct a prompt:

$$P = \pi(x_1, y_1) \oplus \pi(x_2, y_2) \oplus ... \oplus \pi(x_{|\mathcal{T}|}, y_{|\mathcal{T}|}) \oplus \pi(x_{\text{test}}, *) \tag{1}$$

where $\pi$ denotes template-based transformation (see Appendix for illustration) and $\oplus$ denotes concatenation operation. Note that $\pi$ implies a verbalization process that maps label space $Y$ to corresponding tokens $V$ picked from the LM vocabulary. When queried by the prompt $P$, LLM will try to mimic the prepended training examples in the context and predict a probability distribution $p(v|P, \boldsymbol{\theta})$ for the next token $v$. We then map it back to label space $Y$ as prediction:

$$\hat{y}_{\text{test}} = \arg\max_{y \in Y}(v|P, \boldsymbol{\theta}), \quad y \xrightarrow{\pi} v \tag{2}$$

Figure 2 provides a pilot study showing that when prompt $P$ includes more demonstrations, the performance consistently improves, which is in accord with the power law of data scaling investigated in many existing studies (Hestness et al., 2017; Rosenfeld et al., 2020). However, this trend is then prevented by the context length restriction. We provide more comprehensive statistics in Table 1 and Appendix A. In conclusion, this situation poses an actual challenge in many scenarios where one would further collect training examples from few-shots to dozens and expect improved performance, but the power law fails.

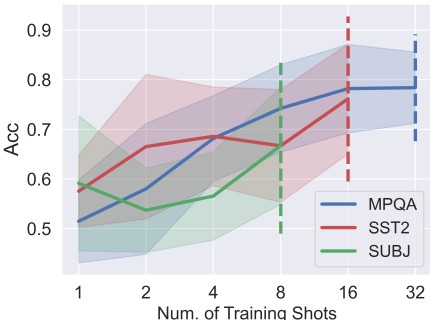

Figure 2: ICL improves with num. of training examples but is limited by context length restriction.

## 3 $k$NN PROMPTING

In this section, we introduce the $k$NN Prompting framework. For a training data set $\mathcal{T}$, we split and exploit it in two respective usage: $\mathcal{T} = \mathcal{D} \cup \mathcal{A}$, i.e., a demonstration set $\mathcal{D} = \{(x_i^d, y_i^d)\}$ and an anchor set $\mathcal{A} = \{(x_i^a, y_i^a)\}$. $k$NN Prompting consists of two stages namely meta test and formal test, the overall framework is illustrated in Figure 3.

**Meta Test**  We build a datastore that caches all anchor examples in $\mathcal{A}$ for later inference time usage. For each $x_i^a$, we respectively concatenate it into prompt $P$, where the prompt prefix is constructed using the demonstration set the same as Equation 1:

$$P_i = \pi(x_1^d, y_1^d) \oplus \pi(x_2^d, y_2^d) \oplus ... \oplus \pi(x_{|\mathcal{D}|}^d, y_{|\mathcal{D}|}^d) \oplus \pi(x_i^a, *) \tag{3}$$

By querying LLM using $P_i$, we obtain the distribution $p(v|P_i, \boldsymbol{\theta})$. Instead of mapping it back to label space $V$, we cache the entire language modeling distribution as the key representation:

$$\boldsymbol{k}_i = p(v|P_i, \boldsymbol{\theta}) \tag{4}$$

Accordingly, label $y_i^a$ is the value. The entire datastore thus consists of paired $\{\boldsymbol{k}_i, y_i^a\}$, we denote the set of keys as $\mathcal{K}$.

**Formal Test**  At inference time, for each test instance $x_{\text{test}}$, we construct the same prompt as Equation 1, and obtain distribution $\boldsymbol{p}_{\text{test}} = p(v|P_{test}, \boldsymbol{\theta})$. We then match the distribution against cached $\mathcal{K}$ in the datastore, where standard $KL$ divergence is used to measure the distance:

$$D_{KL}(\boldsymbol{p}_{\text{test}}||\boldsymbol{k}_i) = \sum_v p(v|P_{test}, \boldsymbol{\theta}) \log \frac{p(v|P_{test}, \boldsymbol{\theta})}{p(v|P_i, \boldsymbol{\theta})} \tag{5}$$

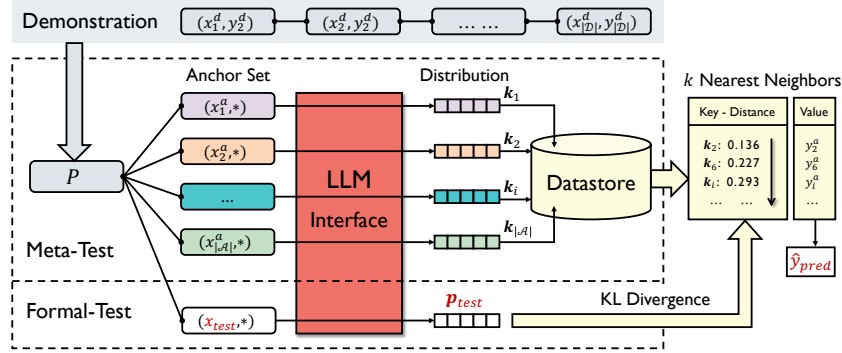

Figure 3: The overall framework of $k$NN Prompting

The predictions are calculated by aggregating its $k$ nearest neighbors:

$$\hat{y}_{pred} = \arg\max_{y \in Y} \sum_{i \in \mathrm{NN}^k(\boldsymbol{p}_{\mathrm{test}}, \mathcal{K})} \mathbb{1}(y_i^a = y) \tag{6}$$

where $\mathrm{NN}^k(*, \mathcal{K})$ denotes the set of $k$ nearest neighbors in $\mathcal{K}$.

## 4 EXPERIMENTS

### 4.1 SETUP

We use 10 established text classification datasets, respectively SST2 (Socher et al., 2013), SUBJ (Pang & Lee, 2004), MPQA (Wiebe et al., 2005), AGNews (Zhang et al., 2015), CB (De Marneffe et al., 2019), CR (Hu & Liu, 2004), DBPedia (Zhang et al., 2015), MR (Pang & Lee, 2005), RTE (Dagan et al., 2005) and TREC (Voorhees & Tice, 2000). For each dataset, we devise intuitive prompt template (Appendix F), and other regarding statistics are listed in Appendix A. We investigate a wide range of LLM scales, including GPT2 (0.8B and 1.5B) (Radford et al., 2019) and the OPT (Zhang et al., 2022) series (3B-30B). GPT2 XL is used for most analyses unless explicitly indicated. We invariantly set the number of neighbors $k$ to 3. There are no other hyper-parameters as the entire framework is training-free. We run with 5 different random seeds for Figure 5 and Figure 6, 10 seeds for all other results. Mean and standard deviation are reported.

### 4.2 DATA UTILITY

In this paper, we refer to data utility as whether and how performance scales up with increased training data. It can be formally depicted by the power law (Hestness et al., 2017):

$$\varepsilon(m) \propto \alpha m^\beta \tag{7}$$

where $m$ is the training data size, $\alpha$ is a constant scaling factor, $\beta$ describes the exponential steepness, and $\varepsilon(*)$ calculates the generalization error, i.e., test time performance. We refer $m$ to training shots, i.e., number of examples per class throughout the paper. In the following sections, we investigate various settings of $m$, respectively few shot ($m \leq M_T$), low resource ($m = 128$), and the overall scaling law ($m \leq 1024$). We refer $M_T$ to the maximum training shots allowed in context for each task $T$, the specific statistics can be found in Appendix A.

### 4.2.1 DATA UTILITY UNDER FEW SHOT SCENARIO

We first investigate $k$NN Prompting under few shot setting, i.e., $m \leq M_T$. We simply set $|\mathcal{D}| = 1$ and use all other examples for $\mathcal{A}$. We leave further exploration of split strategies to Appendix C.5. In order to avoid context length restriction and maintain comparability to baselines as much as possible, 6 out of 10 tasks where $M_T \geq 8$ are selected. The baselines are ICL and state-of-the-art calibration-based augmentations. ContextualCalibration (Zhao et al., 2021) probs the prior bias w.r.t. each category and accordingly calibrate the outputs, and NoisyChannel (Min et al., 2022a) formulates ICL as computing input likelihood conditioned on labels. Results in Table 2 show that

| | Setting&Method | SST2 | SUBJ | MPQA | CR | MR | TREC | AVG |
|---|---|---|---|---|---|---|---|---|
| $m=2$ | ICL | $59.8_{\pm5.9}$ | $51.4_{\pm7.6}$ | $60.2_{\pm14.2}$ | $57.3_{\pm5.5}$ | $64.7_{\pm11.5}$ | $50.0_{\pm3.4}$ | 57.24 |
| | Contextual Calibration | $76.2_{\pm7.0}$ | $69.8_{\pm7.5}$ | $63.4_{\pm10.5}$ | $60.1_{\pm5.7}$ | $75.6_{\pm8.4}$ | $45.5_{\pm7.1}$ | 65.12 |
| | Noisy Channel | $82.6_{\pm3.1}$ | $64.6_{\pm6.0}$ | $60.2_{\pm9.1}$ | $83.3_{\pm2.3}$ | $79.4_{\pm2.1}$ | $35.2_{\pm9.8}$ | **67.54** |
| | $k$NN Prompting | $77.5_{\pm21.3}$ | $73.4_{\pm9.0}$ | $56.6_{\pm20.2}$ | $69.7_{\pm21.3}$ | $81.1_{\pm6.7}$ | $41.3_{\pm12.0}$ | 66.57 |
| | $k$NN Prompting (Partial) | $77.8_{\pm18.9}$ | $68.9_{\pm10.3}$ | $53.3_{\pm22.5}$ | $66.7_{\pm22.7}$ | $81.6_{\pm6.3}$ | $40.2_{\pm8.3}$ | 64.73 |
| $m=4$ | ICL | $67.2_{\pm12.6}$ | $56.5_{\pm11.8}$ | $70.8_{\pm11.4}$ | $60.8_{\pm11.7}$ | $62.8_{\pm10.3}$ | $50.9_{\pm4.3}$ | 61.50 |
| | Contextual Calibration | $70.8_{\pm11.2}$ | $60.0_{\pm8.1}$ | $70.5_{\pm9.0}$ | $59.6_{\pm6.7}$ | $70.0_{\pm7.0}$ | $43.6_{\pm4.5}$ | 62.43 |
| | Noisy Channel | $80.9_{\pm3.1}$ | $60.5_{\pm8.7}$ | $66.4_{\pm6.0}$ | $83.9_{\pm2.3}$ | $79.0_{\pm2.9}$ | $40.9_{\pm9.3}$ | 68.58 |
| | $k$NN Prompting | $87.1_{\pm6.2}$ | $73.5_{\pm7.7}$ | $66.4_{\pm11.7}$ | $71.2_{\pm17.7}$ | $82.9_{\pm3.0}$ | $51.6_{\pm11.2}$ | **72.14** |
| | $k$NN Prompting (Partial) | $85.9_{\pm6.7}$ | $70.5_{\pm10.3}$ | $67.4_{\pm12.0}$ | $68.1_{\pm20.2}$ | $82.0_{\pm3.6}$ | $52.2_{\pm7.9}$ | 71.00 |
| $m=8$ | ICL | $57.8_{\pm5.6}$ | $66.2_{\pm13.0}$ | $77.2_{\pm11.0}$ | $66.0_{\pm11.5}$ | $61.5_{\pm6.5}$ | $50.9_{\pm6.1}$ | 63.27 |
| | Contextual Calibration | $68.5_{\pm7.9}$ | $64.5_{\pm9.9}$ | $72.7_{\pm11.1}$ | $64.9_{\pm6.8}$ | $68.2_{\pm8.8}$ | $44.0_{\pm5.2}$ | 63.80 |
| | Noisy Channel | $82.0_{\pm2.1}$ | $62.5_{\pm5.7}$ | $70.1_{\pm4.2}$ | $85.0_{\pm2.1}$ | $79.2_{\pm2.1}$ | $41.6_{\pm9.5}$ | 70.06 |
| | $k$NN Prompting | $88.9_{\pm2.1}$ | $77.7_{\pm5.8}$ | $72.5_{\pm12.4}$ | $75.4_{\pm12.1}$ | $84.6_{\pm1.7}$ | $63.7_{\pm5.5}$ | **77.13** |
| | $k$NN Prompting (Partial) | $88.9_{\pm2.3}$ | $69.2_{\pm11.1}$ | $67.8_{\pm15.9}$ | $72.8_{\pm12.7}$ | $84.3_{\pm2.2}$ | $54.6_{\pm3.7}$ | 72.92 |
| $m=16$ | ICL | $67.7_{\pm10.8}$ | $75.5^{\dagger}_{\pm11.4}$ | $77.6_{\pm7.6}$ | $73.3^{\dagger}_{\pm11.9}$ | $61.8^{\dagger}_{\pm5.6}$ | $52.0^{\dagger}_{\pm5.1}$ | 67.97 |
| | Contextual Calibration | $75.7_{\pm7.1}$ | $59.7^{\dagger}_{\pm6.5}$ | $75.2_{\pm8.0}$ | $73.0^{\dagger}_{\pm7.6}$ | $73.1^{\dagger}_{\pm7.0}$ | $46.5^{\dagger}_{\pm6.2}$ | 67.21 |
| | Noisy Channel | $84.4_{\pm1.4}$ | $62.4^{\dagger}_{\pm7.2}$ | $70.4_{\pm6.2}$ | $83.7^{\dagger}_{\pm3.3}$ | $79.6^{\dagger}_{\pm2.7}$ | $54.2^{\dagger}_{\pm7.8}$ | 72.44 |
| | $k$NN Prompting | $88.8_{\pm1.6}$ | $80.9_{\pm4.0}$ | $68.2_{\pm7.6}$ | $80.1_{\pm4.7}$ | $84.8_{\pm2.7}$ | $70.0_{\pm3.9}$ | **78.80** |
| | $k$NN Prompting (Partial) | $89.7_{\pm2.5}$ | $71.4_{\pm9.8}$ | $60.5_{\pm12.3}$ | $79.8_{\pm5.4}$ | $84.8_{\pm3.0}$ | $55.8_{\pm4.0}$ | 73.67 |
| $m=32$ | ICL | $66.5^{\dagger}_{\pm10.4}$ | $70.1^{\dagger}_{\pm9.6}$ | $77.4_{\pm8.2}$ | $67.7^{\dagger}_{\pm9.6}$ | $63.6^{\dagger}_{\pm8.7}$ | $52.0^{\dagger}_{\pm2.1}$ | 66.22 |
| | Contextual Calibration | $76.9^{\dagger}_{\pm7.5}$ | $58.6^{\dagger}_{\pm9.2}$ | $76.5_{\pm7.9}$ | $78.5^{\dagger}_{\pm7.8}$ | $71.2^{\dagger}_{\pm7.4}$ | $44.3^{\dagger}_{\pm3.2}$ | 67.66 |
| | Noisy Channel | $84.8^{\dagger}_{\pm0.9}$ | $61.1^{\dagger}_{\pm3.9}$ | $70.8_{\pm5.2}$ | $82.5^{\dagger}_{\pm2.4}$ | $80.0^{\dagger}_{\pm1.8}$ | $47.6^{\dagger}_{\pm9.1}$ | 71.14 |
| | $k$NN Prompting | $89.0_{\pm1.9}$ | $83.2_{\pm3.9}$ | $69.3_{\pm7.9}$ | $77.8_{\pm5.6}$ | $85.0_{\pm2.0}$ | $73.5_{\pm3.9}$ | **79.64** |
| | $k$NN Prompting (Partial) | $86.7_{\pm5.0}$ | $65.9_{\pm12.9}$ | $64.4_{\pm12.1}$ | $74.2_{\pm7.4}$ | $83.9_{\pm3.8}$ | $59.4_{\pm4.0}$ | 72.41 |

Table 2: Results under few-shot scenario. Calibration-based baselines are reproduced using their released code[34]. $\dagger$ denotes necessary truncation. The overall scaling trend is accordingly visualized in Figure 4 and Appendix C.1. **Partial** means only label-words distribution is utilized.

$k$NN Prompting significantly outperforms competitive baselines under strictly comparable settings ($m \leq 8$), specifically, **+3.56** for 4 shot, and **+7.07** for 8 shot.

**Superiority of Whole LM Distribution** Calibration-based methods (Zhao et al., 2021) as well as standard ICL only access label words instead of whole LM distribution, which is inferior in two aspects: 1) loss of information. LLM always generate distribution over all words in the entire vocabulary, non-label words probabilities also reflect its understanding in certain perspectives; 2) multiple label words competing with each other. There exist various choices for label words but no oracle rules to select one, and alternative choices potentially compete with the selected label words, distorting the label space distribution. This is also referred to as surface form competition (Holtzman et al., 2021). In Table 2 we very this benefit by masking out non-label words (referred to as **Partial**).

### 4.2.2 DATA UTILITY BEYOND THE CONTEXT

We then investigate a major advantage of $k$NN Prompting, which is scaling up to more training examples that otherwise would not fit in the context. We increase $m$ to 128 and compare with: 1) ICL Ensemble which is an intuitive alternative to scale ICL up, and has been adopted in previous works (Jiang et al., 2020); 2) finetuning of standard PLM such as BERT or GPT Large, which could produce meaningful results with such amount of data. For all methods, we append maximum $M_T$ examples into prompt $P$. For ICL Ensemble, we split $\mathcal{T}$ into multiple non-overlap demonstration sets $\mathcal{T} = \mathcal{D}_1 \cup \mathcal{D}_2 \cup ... \cup \mathcal{D}_N$ to construct different prompts, and ensemble their predictions.

Results in Table 3 show that $k$NN Prompting continues to improve and outperform ICL and its ensemble respectively by **+16.96** and **+16.08** (0.8B model, average score). Besides, the ensemble baseline is also very inefficient, assuming $M_T = 8$, we need to query $128/8 = 16$ times for every test instance, and this keeps growing linearly if we use more training data, eventually becomes prohibitively inefficient and can not scale up either. $k$NN Prompting also outperforms FT when the adopted LLM scales above **6B** (82.73). By contrast, it would require LLM to scale above **30B** (82.45) using ICL Ensemble and even larger using standard ICL.

---

[3] https://github.com/tonyzhaozh/few-shot-learning

[4] https://github.com/shmsw25/Channel-LM-Prompting, as there exists slightly difference of prompt templates, we report their best template out of four.

| Models & Methods | | SST2 | SUBJ | MPQA | AGNews | CB | CR | DBPedia | MR | RTE | TREC | AVG |
|---|---|---|---|---|---|---|---|---|---|---|---|---|
| BERT Large FT | | $88.3_{\pm1.4}$ | $90.7_{\pm0.6}$ | $74.5_{\pm5.0}$ | $88.0_{\pm1.0}$ | $78.6_{\pm3.6}$ | $88.0_{\pm2.6}$ | $95.1_{\pm1.8}$ | $83.0_{\pm3.1}$ | $58.1_{\pm1.5}$ | $78.8_{\pm5.3}$ | 82.31 |
| GPT Large FT | | $90.7_{\pm1.3}$ | $86.1_{\pm1.7}$ | $87.6_{\pm0.9}$ | $88.3_{\pm1.5}$ | $70.0_{\pm2.0}$ | $86.7_{\pm13.2}$ | $96.5_{\pm1.2}$ | $86.2_{\pm1.0}$ | $55.4_{\pm3.8}$ | $71.2_{\pm2.2}$ | 81.88 |
| 0.8B | ICL | $63.4_{\pm7.3}$ | $58.9_{\pm8.7}$ | $70.5_{\pm5.2}$ | $61.7_{\pm15.4}$ | $45.0_{\pm9.1}$ | $83.3_{\pm13.7}$ | $59.9_{\pm11.5}$ | $77.0_{\pm15.7}$ | $53.6_{\pm3.1}$ | $54.4_{\pm1.7}$ | 62.77 |
| | ICL Ensemble | $63.0_{\pm6.4}$ | $57.7_{\pm10.3}$ | $69.1_{\pm6.2}$ | $67.4_{\pm2.9}$ | $41.1_{\pm3.1}$ | $83.8_{\pm11.5}$ | $67.8_{\pm3.7}$ | $72.7_{\pm12.3}$ | $55.1_{\pm3.8}$ | $59.0_{\pm3.7}$ | 63.65 |
| | kNN Prompting | $84.5_{\pm5.3}$ | $85.8_{\pm1.6}$ | $83.1_{\pm0.8}$ | $84.5_{\pm1.3}$ | $62.1_{\pm3.4}$ | $89.7_{\pm0.6}$ | $95.8_{\pm0.5}$ | $84.0_{\pm1.8}$ | $53.6_{\pm3.2}$ | $74.2_{\pm4.4}$ | **79.73** |
| 1.5B | ICL | $81.3_{\pm5.4}$ | $64.1_{\pm11.3}$ | $75.2_{\pm8.8}$ | $72.7_{\pm18.5}$ | $60.7_{\pm2.8}$ | $66.2_{\pm16.7}$ | $83.5_{\pm3.8}$ | $72.2_{\pm13.9}$ | $53.0_{\pm1.7}$ | $54.2_{\pm4.9}$ | 68.31 |
| | ICL Ensemble | $83.4_{\pm1.9}$ | $63.4_{\pm11.5}$ | $75.0_{\pm7.2}$ | $81.1_{\pm0.7}$ | $52.9_{\pm8.1}$ | $63.4_{\pm17.2}$ | $83.7_{\pm1.3}$ | $73.8_{\pm15.1}$ | $55.9_{\pm1.8}$ | $59.6_{\pm2.3}$ | 69.21 |
| | kNN Prompting | $86.3_{\pm2.9}$ | $83.8_{\pm2.1}$ | $82.3_{\pm1.5}$ | $87.2_{\pm0.4}$ | $64.6_{\pm7.9}$ | $88.9_{\pm2.4}$ | $96.5_{\pm0.7}$ | $86.4_{\pm0.8}$ | $51.1_{\pm1.7}$ | $74.0_{\pm2.9}$ | **80.12** |
| 2.7B | ICL | $89.9_{\pm4.5}$ | $77.7_{\pm5.8}$ | $84.5_{\pm2.1}$ | $78.8_{\pm4.0}$ | $51.1_{\pm6.1}$ | $92.6_{\pm0.6}$ | $88.8_{\pm1.3}$ | $92.5_{\pm1.0}$ | $52.4_{\pm4.2}$ | $64.6_{\pm3.0}$ | 77.29 |
| | ICL Ensemble | $90.2_{\pm3.7}$ | $77.7_{\pm3.4}$ | $86.6_{\pm1.9}$ | $78.4_{\pm1.8}$ | $54.6_{\pm1.0}$ | $93.4_{\pm1.1}$ | $89.3_{\pm0.9}$ | $92.4_{\pm0.8}$ | $54.3_{\pm2.3}$ | $78.8_{\pm2.3}$ | 79.57 |
| | kNN Prompting | $93.4_{\pm1.3}$ | $87.5_{\pm2.0}$ | $83.3_{\pm3.9}$ | $86.7_{\pm2.4}$ | $57.1_{\pm3.8}$ | $90.2_{\pm2.0}$ | $98.6_{\pm0.6}$ | $91.4_{\pm1.5}$ | $53.4_{\pm5.6}$ | $80.8_{\pm2.0}$ | **82.25** |
| 6B | ICL | $92.7_{\pm1.8}$ | $81.6_{\pm4.6}$ | $86.2_{\pm1.9}$ | $68.8_{\pm8.2}$ | $52.5_{\pm9.7}$ | $92.0_{\pm2.6}$ | $89.8_{\pm0.9}$ | $91.6_{\pm1.1}$ | $55.0_{\pm1.6}$ | $62.0_{\pm7.3}$ | 77.19 |
| | ICL Ensemble | $93.2_{\pm0.9}$ | $85.6_{\pm3.8}$ | $87.7_{\pm2.9}$ | $71.1_{\pm5.5}$ | $45.7_{\pm2.0}$ | $93.4_{\pm1.5}$ | $90.9_{\pm1.0}$ | $92.0_{\pm0.2}$ | $54.8_{\pm1.3}$ | $70.3_{\pm1.8}$ | 78.47 |
| | kNN Prompting | $92.2_{\pm1.2}$ | $87.4_{\pm1.7}$ | $85.2_{\pm2.5}$ | $87.4_{\pm1.6}$ | $63.6_{\pm4.1}$ | $90.6_{\pm2.2}$ | $98.5_{\pm0.4}$ | $91.0_{\pm1.5}$ | $55.6_{\pm2.0}$ | $75.8_{\pm3.1}$ | **82.73** |
| 13B | ICL | $89.0_{\pm4.3}$ | $91.3_{\pm2.4}$ | $78.4_{\pm7.2}$ | $78.1_{\pm5.6}$ | $53.2_{\pm4.4}$ | $93.4_{\pm1.1}$ | $92.2_{\pm2.4}$ | $89.9_{\pm2.2}$ | $55.8_{\pm3.0}$ | $55.5_{\pm6.1}$ | 77.68 |
| | ICL Ensemble | $88.2_{\pm4.7}$ | $90.7_{\pm1.2}$ | $78.4_{\pm4.8}$ | $82.6_{\pm1.5}$ | $62.1_{\pm3.9}$ | $94.0_{\pm0.7}$ | $94.6_{\pm1.0}$ | $89.0_{\pm2.5}$ | $56.2_{\pm2.0}$ | $59.8_{\pm3.7}$ | 79.56 |
| | kNN Prompting | $94.8_{\pm0.6}$ | $90.1_{\pm1.5}$ | $86.2_{\pm3.8}$ | $87.4_{\pm1.7}$ | $78.9_{\pm4.8}$ | $89.6_{\pm1.1}$ | $98.9_{\pm0.5}$ | $92.0_{\pm0.5}$ | $58.2_{\pm4.4}$ | $73.1_{\pm2.9}$ | **84.93** |
| 30B | ICL | $90.8_{\pm4.1}$ | $83.5_{\pm8.7}$ | $80.7_{\pm1.9}$ | $74.8_{\pm4.6}$ | $64.6_{\pm8.3}$ | $87.7_{\pm3.9}$ | $93.3_{\pm0.4}$ | $93.4_{\pm1.0}$ | $61.6_{\pm2.7}$ | $71.7_{\pm2.7}$ | 80.21 |
| | ICL Ensemble | $92.6_{\pm2.3}$ | $84.1_{\pm4.6}$ | $79.9_{\pm1.9}$ | $78.3_{\pm2.4}$ | $67.1_{\pm4.8}$ | $88.1_{\pm2.9}$ | $93.8_{\pm1.0}$ | $93.4_{\pm1.0}$ | $65.1_{\pm4.6}$ | $82.0_{\pm2.5}$ | 82.45 |
| | kNN Prompting | $94.3_{\pm0.9}$ | $92.7_{\pm1.7}$ | $84.5_{\pm0.9}$ | $87.1_{\pm1.4}$ | $70.7_{\pm8.2}$ | $91.0_{\pm1.2}$ | $98.8_{\pm0.5}$ | $93.1_{\pm1.5}$ | $61.7_{\pm4.8}$ | $79.8_{\pm2.0}$ | **85.38** |

Table 3: Results under low resource scenario ($m = 128$). Compared with FT and ICL Ensemble.

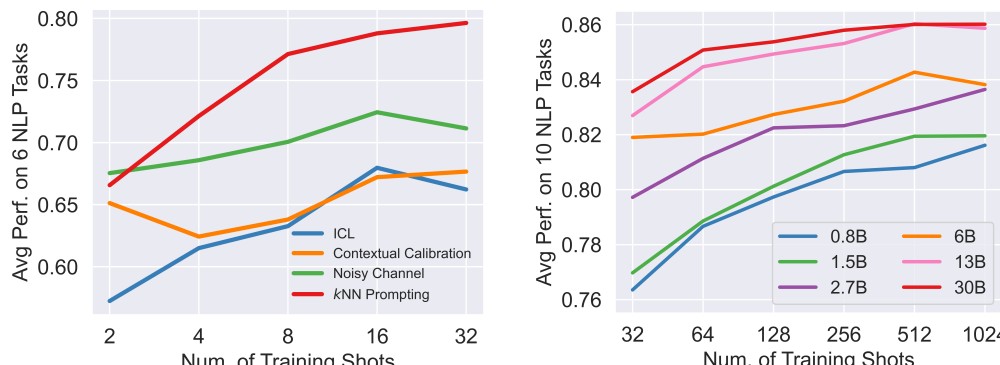

Figure 4: Data scaling under few shot scenario. Compared with calibration-based methods.

Figure 5: Data scaling under fully supervised scenario. Conducted across various LLM scales.

### 4.2.3 Continually Scaling Up to Thousands of Training Data

We now fully scale data up to thousands of training examples and provide extensive results across model scales to observe their overall scaling performance. Figure 5 shows kNN prompting can continually generalize across the tested range to provide effective data utility, re-enabling the power law under gradient-free paradigm of LLM deployment. The full results can be found in Appendix C.2. With only 32 shots training data, kNN prompting dramatically improves ICL by **+13.58** in average score at its most (0.8 B), and achieves absolute improvements up to **+18.84** under fully supervised setting. With the largest model OPT 30B, it achieves a best performance of 86.02.

**Comparison to Demonstration Selection** A line of related works try to utilize available training data by firstly retrieving the most relevant ones from the entire training set, then selectively composing the prompt before querying LLM (Liu et al., 2022b; Rubin et al., 2022). We since reproduce such methods according to Liu et al. (2022b). We employ state-of-the-art general-purpose sentence encoders[5] to represent test and training instance, and compute their cosine similarity, the most similar $M_T$ examples are selected to construct prompt $P$. These retrieving models include BM25 (Trotman et al., 2014), Sentence-BERT (Reimers & Gurevych, 2019), SimCSE (Gao et al., 2021) and Trans-Encoder (Liu et al., 2022a).

Figure 6 shows that although such methods indeed exhibits marginal scaling benefits, they are nowhere near competitive against kNN Prompting. Full results are listed in Appendix C.3. To fur-

---

[5]Note that Liu et al. (2022b) also employ RoBERTa model finetuned on SST2, but this overlaps with our selected benchmark and does not generalize to other tasks beyond sentiment classification.

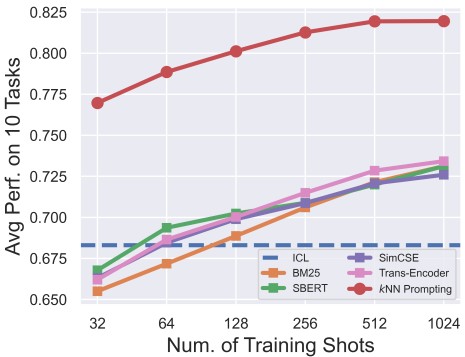

Figure 6: cf. Demonstration Selection.

| Method | SST2 | SUBJ | MPQA |
|---|---|---|---|
| ICL Baseline | $81.3_{\pm 5.4}$ | $64.1_{\pm 11.3}$ | $75.2_{\pm 8.8}$ |
| **DemonSelection (upper)** | **92.6** (1‰) | 86.0 (3‰) | **87.5** (1‰) |
| $k$NN Prompting | $88.2_{\pm 1.0}$ | $\mathbf{88.4}_{\pm 1.8}$ | $84.1_{\pm 1.2}$ |
| | **MR** | **CR** | **TREC** |
| ICL Baseline | $72.2_{\pm 13.9}$ | $66.2_{\pm 16.7}$ | $54.2_{\pm 4.9}$ |
| **DemonSelection (upper)** | **88.7** (1‰) | **88.7** (1‰) | 73.0 (1‰) |
| $k$NN Prompting | $84.4_{\pm 1.5}$ | $86.7_{\pm 1.3}$ | $\mathbf{83.0}_{\pm 1.4}$ |

Table 4: Comparison to **upper-bound** of Demonstration Selection. 1‰ inside the parentheses means the result can be achieved in 1 runs out of 1,000 searches.

ther solidify this conclusion, we push PromptCompose to an extreme situation trying to approximate its upper-bound. As such methods ultimately resort to compose the prompt, it should be bounded by the best composition scheme from finite compositions. We thus search for 1,000 prompts with different examples and report their best run. In Table 4 we find $k$NN Prompting performs on par with or even surpasses such upper-bound approximation. In conclusion, $k$NN Prompting essentially makes better use of training examples than PromptCompose as the latter still only refer to in-context examples during LLM inference while most training data are discarded beforehand.

### 4.3 ANALYSES AND EXPLANATION

#### 4.3.1 ROBUSTNESS W.R.T. DIFFERENT TRAINING EXAMPLES

It is previously found that vanilla ICL suffers from severe instability (Zhao et al., 2021). Figure 7 is produced with 10 different seeds, which results in different choices and permutations of training data. We show that $k$NN Prompting significantly improves the robustness. Besides, the performance becomes more robust with increasing anchor set size. On 10 investigated tasks, the standard deviation of $k$NN Prompting (3.83) is significantly smaller than ICL (9.14)[6].

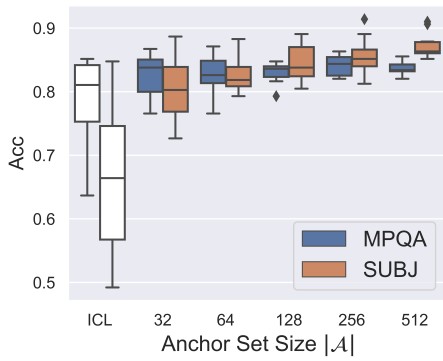

Figure 7: Robustness.

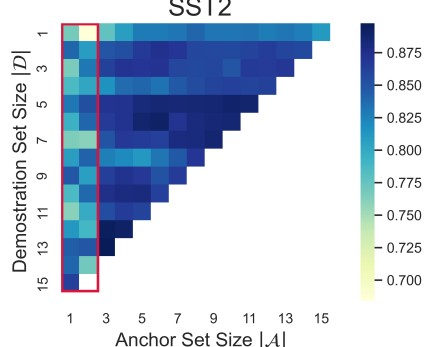

Figure 8: Split strategy.

#### 4.3.2 SPLIT STRATEGY BETWEEN DEMONSTRATIONS AND ANCHORS

For fully supervised scenario where $m \gg M_T$, we can simply set $|\mathcal{D}|$ to its maximum $M_T$. Otherwise, we might need to deliberate the trade-off between the demonstration set and anchor set. In Figure 13 we search through all possible combinations given that $|\mathcal{A}| + |\mathcal{D}| \leq M_T$. We see the left part of the heatmap generally performs inferior, i.e., $|\mathcal{A}| \in \{1, 2\}$. This means a larger $|\mathcal{A}|$ contributes more to the performance while the choice for $|\mathcal{D}|$ is relatively more robust. The conclusion also corresponds to few shot results that anchor set yields better data utility than context concatenation. More datasets are also provided in Appendix C.5.

---

[6]Calculated using Table 8 statistics with 32 shot setting, 0.8B model

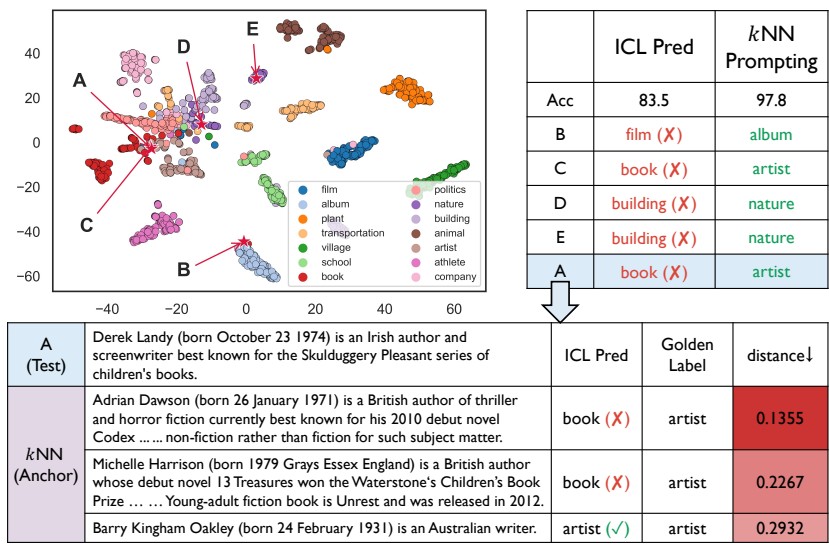

Figure 9: t-SNE (van der Maaten & Hinton, 2008) for anchors and test cases. Cases are randomly selected given that $k$NN Prompting outperforms ICL. DBPedia is an ontology classification task.

### 4.3.3 QUALITATIVE ANALYSES AND REASONS OF EFFECTIVENESS

We first formally organize the explanation as follows according to Figure 9:

- The output language modeling (LM) distribution of LLM is essentially not well aligned with task-specific label space, resulting in inferior performance (*83.5 test accuracy*) of default ICL.

- If we similarly perform inference on anchor set, we would expect to get approximately *83.5 anchor accuracy* by assuming i.i.d. data distribution. However, we are already aware of each of their golden labels, which actually gives *100 anchor accuracy*.

- LM distribution is inferior for making direct predictions, but superior for matching examples because it entails distributional, delicate and comprehensive representations generated by LLM. $k$NN Prompting leverages such representations (*83.5 accuracy*) only for matching, and refer to their golden labels (*100 accuracy*) for predicting, thus successfully transfer part of the knowledge originating from anchor labels to test instances.

In the visualization, the representations generally exhibit partially clustered pattern, we can identify proportional examples that get entangled with different categories and crowded together (Case A, C, D), these confusing cases are likely to cause erroneous predictions in ICL and corresponds to under-performed 83.5 accuracy as mentioned above. Specifically, case $A$ is an abstract about a novelist and should belong to category **artist**, but it is easily confused with category **book** using ICL because the context did mention books. By contrast, $k$NN Prompting can correctly predict by referring to similar anchors that are also about novelists and their books (as listed in the table). Some of the anchors are also incorrectly predicted as **book**, but it no longer matters because $k$NN Prompting only use the distribution for nearest neighbor search but refer to golden labels for prediction. Besides, as we can clearly know how the prediction is made, i.e., which anchor examples are referred, the proposed method also exhibits explainability as a further advantage.

## 5 RELATED WORKS

Large language models, since firstly scaled up to hundreds of billions parameters by Brown et al. (2020) and followed by several others (Rae et al., 2021; Zhang et al., 2022; Chowdhery et al., 2022; Cohen et al., 2022; Smith et al., 2022), have become the most prominent direction of NLP area. Although these models exhibit surprisingly powerful and even emergent capabilities in a wide range of NLP tasks (Wei et al., 2022b; Hendrycks et al., 2020; Srivastava et al., 2022), they are prohibitively expensive for most researchers or users to train or even hold. In-Context Learning, which suits LLM to various tasks while requires no training, therefore becomes the typical usage as

is popularized by Brown et al. (2020). Similar ideas of formulating target tasks into natural language sequences can also be found in earlier works (Trinh & Le, 2018; Raffel et al., 2020).

To better exploit LLM for various scenarios, it becomes a crucial problem to develop augmented methods for ICL (Dong et al., 2022). Xie et al. (2022) provide theoretical explanations that formalize ICL as Bayesian inference. Dai et al. (2022) reveal that ICL can be seen as implicit finetuning where LLM produces meta-gradients from in-context demonstrations to adapt the model behavior. Wei et al. (2022a) and Sanh et al. (2022) propose instruction tuning, which further pretrains LLM with a collection of downstream tasks in a shared prompting format. Min et al. (2022b) and Chen et al. (2022b) introduce meta-learning to better adapt LMs to ICL. Wei et al. (2022c) and Kojima et al. (2022) propose to augment the demonstrations with human-aided reasoning steps or hints, which surprisingly improved the performance for arithmetics and other reasoning tasks. Closely related to this work, Liu et al. (2022b) and Rubin et al. (2022) propose to compose prompt $P$ by selecting most similar training examples. Zhao et al. (2021) and Min et al. (2022a) propose to calibrate ICL prediction via either probing the bias or reversing the conditional prediction formulation.

$k$NN is a classical machine learning algorithm (Fix & Hodges, 1989) well known for its simplicity and inspired a wide range of application (Papernot & McDaniel, 2018; Orhan, 2018). In the field of NLP, Kaiser et al. (2017) construct a differentiable memory module for nearest neighbor searching which improves generalization to rare events. Similar idea has also been explored for generation tasks (Guu et al., 2018), such as dialog generation (Weston et al., 2018), machine translation (Khandelwal et al., 2021), etc. Wang et al. (2022) and Chen et al. (2022a) propose to retrieve similar training examples and incorporate them into the input to jointly train the model. While Khandelwal et al. (2020) and Shi et al. (2022) consider unsupervised corpus as datastore, retrieve and interpolate them with the current step language modeling probability. The retrieved corpus can also serve as references for knowledge intensive tasks, but the retriever would need explicitly training for such purpose (Lewis et al., 2020; Borgeaud et al., 2022; Izacard et al., 2022). Different from these works, $k$NN Prompting is suitably situated in the gradient-free paradigm of LLM deployment, which avoids calibration treatment and effectively bridges data scaling into model scaling.

## 6 DISCUSSION

Under the existing ICL paradigm, it is often impossible to take advantage of both the capability of LLM and the data utility of finetuning, i.e., model scaling and data scaling. $k$NN Prompting finds an effective solution to promise them both. Nevertheless, we assume its data utility should still be inferior to the specialized finetuning of LLMs, if given sufficient computation resources in an ideal setting. We believe that it is a very important and promising direction to further approach this upper-bound and expect to raise more interests in future works.

A potential concern for retrieval-based models is their efficiency, especially when corpus level datastore is utilized. $k$NN Prompting is free of such concerns as it considers training data, which is in manageable scale. Under few shot scenario, $k$NN Prompting even reduces computational costs at deployment time compared to standard ICL. It works well with one shot demonstrations as experimented in Section 4.2.1, while the anchor examples are queried for only once and cached locally. By contrast, existing methods only perform better when all examples are prepended in a single prompt. This advantage is rather important as we need to repeatedly query the prompt in practical usage of LLM service, and longer prompt results in linearly more monetary costs if charged by token numbers or super-linearly more computational costs if measured by FLOPS.

## 7 CONCLUSION

In this paper, we propose $k$NN Prompting as a simple and effective solution to advance gradient-free deployment of LLM inference. Motivated as calibration-free optimization, $k$NN Prompting significantly outperforms state-of-the-art calibration-based methods under comparable few shot scenario. While its major advantage is further revealed when training data increases and can not fit in the context. $k$NN Prompting can effectively scale up with as many training data as are available, successfully bridging the utility of data scaling into model scaling. The proposed framework endeavors to realize more effective, efficient and applicable utilization of large language models in realistic scenarios, and hopefully could inspire further research interests toward the same goal.

Eᴛʜɪᴄᴀʟ Cᴏɴsɪᴅᴇʀᴀᴛɪᴏɴs

This work is built upon the ICL paradigm and involves querying LLM for responses. These models might generate contents with potential ethical risks regarding fairness and bias (Bommasani et al., 2021; Blodgett et al., 2020), depending on specific downstream tasks. Although the scope of this paper remains on how to better exploit LLM for task performance, it is worth further discussion to combine the proposed framework in conjunction with well-established methods that can measure (Nadeem et al., 2021) and mitigate (Nadeem et al., 2021; Gupta et al., 2022) such ethical risks.

Aᴄᴋɴᴏᴡʟᴇᴅɢᴍᴇɴᴛs

This work is supported in part by the National Natural Science Foundation of China under Grant 62222212, 62232006 and 61876223, Science Fund for Creative Research Groups under Grant 62121002.

Rᴇғᴇʀᴇɴᴄᴇs

Su Lin Blodgett, Solon Barocas, Hal Daumé III, and Hanna Wallach. Language (technology) is power: A critical survey of "bias" in NLP. In *Proceedings of the 58th Annual Meeting of the Association for Computational Linguistics*, pp. 5454–5476, Online, July 2020. Association for Computational Linguistics. doi: 10.18653/v1/2020.acl-main.485. URL `https://aclanthology.org/2020.acl-main.485`.

Rishi Bommasani, Drew A. Hudson, Ehsan Adeli, Russ Altman, Simran Arora, Sydney von Arx, Michael S. Bernstein, Jeannette Bohg, Antoine Bosselut, Emma Brunskill, Erik Brynjolfsson, Shyamal Buch, Dallas Card, Rodrigo Castellon, Niladri S. Chatterji, Annie S. Chen, Kathleen Creel, Jared Quincy Davis, Dorottya Demszky, Chris Donahue, Moussa Doumbouya, Esin Durmus, Stefano Ermon, John Etchemendy, Kawin Ethayarajh, Li Fei-Fei, Chelsea Finn, Trevor Gale, Lauren Gillespie, Karan Goel, Noah D. Goodman, Shelby Grossman, Neel Guha, Tatsunori Hashimoto, Peter Henderson, John Hewitt, Daniel E. Ho, Jenny Hong, Kyle Hsu, Jing Huang, Thomas Icard, Saahil Jain, Dan Jurafsky, Pratyusha Kalluri, Siddharth Karamcheti, Geoff Keeling, Fereshte Khani, Omar Khattab, Pang Wei Koh, Mark S. Krass, Ranjay Krishna, Rohith Kuditipudi, and et al. On the opportunities and risks of foundation models. *CoRR*, abs/2108.07258, 2021. URL `https://arxiv.org/abs/2108.07258`.

Sebastian Borgeaud, Arthur Mensch, Jordan Hoffmann, Trevor Cai, Eliza Rutherford, Katie Millican, George Bm Van Den Driessche, Jean-Baptiste Lespiau, Bogdan Damoc, Aidan Clark, Diego De Las Casas, Aurelia Guy, Jacob Menick, Roman Ring, Tom Hennigan, Saffron Huang, Loren Maggiore, Chris Jones, Albin Cassirer, Andy Brock, Michela Paganini, Geoffrey Irving, Oriol Vinyals, Simon Osindero, Karen Simonyan, Jack Rae, Erich Elsen, and Laurent Sifre. Improving language models by retrieving from trillions of tokens. In Kamalika Chaudhuri, Stefanie Jegelka, Le Song, Csaba Szepesvari, Gang Niu, and Sivan Sabato (eds.), *Proceedings of the 39th International Conference on Machine Learning*, volume 162 of *Proceedings of Machine Learning Research*, pp. 2206–2240. PMLR, 17–23 Jul 2022. URL `https://proceedings.mlr.press/v162/borgeaud22a.html`.

Tom Brown, Benjamin Mann, Nick Ryder, Melanie Subbiah, Jared D Kaplan, Prafulla Dhariwal, Arvind Neelakantan, Pranav Shyam, Girish Sastry, Amanda Askell, Sandhini Agarwal, Ariel Herbert-Voss, Gretchen Krueger, Tom Henighan, Rewon Child, Aditya Ramesh, Daniel Ziegler, Jeffrey Wu, Clemens Winter, Chris Hesse, Mark Chen, Eric Sigler, Mateusz Litwin, Scott Gray, Benjamin Chess, Jack Clark, Christopher Berner, Sam McCandlish, Alec Radford, Ilya Sutskever, and Dario Amodei. Language models are few-shot learners. In H. Larochelle, M. Ranzato, R. Hadsell, M.F. Balcan, and H. Lin (eds.), *Advances in Neural Information Processing Systems*, volume 33, pp. 1877–1901. Curran Associates, Inc., 2020. URL `https://proceedings.neurips.cc/paper/2020/file/1457c0d6bfcb4967418bfb8ac142f64a-Paper.pdf`.

Xiang Chen, Lei Li, Ningyu Zhang, Xiaozhuan Liang, Shumin Deng, Chuanqi Tan, Fei Huang, Luo Si, and Huajun Chen. Decoupling knowledge from memorization: Retrieval-augmented prompt learning. *arXiv preprint arXiv:2205.14704*, 2022a.

Yanda Chen, Ruiqi Zhong, Sheng Zha, George Karypis, and He He. Meta-learning via language model in-context tuning. In *Proceedings of the 60th Annual Meeting of the Association for Computational Linguistics (Volume 1: Long Papers)*, pp. 719–730, Dublin, Ireland, May 2022b. Association for Computational Linguistics. doi: 10.18653/v1/2022.acl-long.53. URL `https://aclanthology.org/2022.acl-long.53`.

Aakanksha Chowdhery, Sharan Narang, Jacob Devlin, Maarten Bosma, Gaurav Mishra, Adam Roberts, Paul Barham, Hyung Won Chung, Charles Sutton, Sebastian Gehrmann, et al. Palm: Scaling language modeling with pathways. *arXiv preprint arXiv:2204.02311*, 2022.

Aaron Daniel Cohen, Adam Roberts, Alejandra Molina, Alena Butryna, Alicia Jin, Apoorv Kulshreshtha, Ben Hutchinson, Ben Zevenbergen, Blaise Hilary Aguera-Arcas, Chung-ching Chang, et al. Lamda: Language models for dialog applications. 2022.

Ido Dagan, Oren Glickman, and Bernardo Magnini. The pascal recognising textual entailment challenge. In *Proceedings of the First International Conference on Machine Learning Challenges: Evaluating Predictive Uncertainty Visual Object Classification, and Recognizing Textual Entailment*, MLCW'05, pp. 177–190, Berlin, Heidelberg, 2005. Springer-Verlag. ISBN 3540334270. doi: 10.1007/11736790_9. URL `https://doi.org/10.1007/11736790_9`.

Damai Dai, Yutao Sun, Li Dong, Yaru Hao, Zhifang Sui, and Furu Wei. Why can gpt learn in-context? language models secretly perform gradient descent as meta optimizers. *arXiv preprint arXiv:2212.10559*, 2022.

Marie-Catherine De Marneffe, Mandy Simons, and Judith Tonhauser. The commitmentbank: Investigating projection in naturally occurring discourse. In *proceedings of Sinn und Bedeutung*, volume 23, pp. 107–124, 2019.

Qingxiu Dong, Lei Li, Damai Dai, Ce Zheng, Zhiyong Wu, Baobao Chang, Xu Sun, Jingjing Xu, and Zhifang Sui. A survey for in-context learning. *arXiv preprint arXiv:2301.00234*, 2022.

Evelyn Fix and Joseph Lawson Hodges. Discriminatory analysis. nonparametric discrimination: Consistency properties. *International Statistical Review/Revue Internationale de Statistique*, 57 (3):238–247, 1989.

Tianyu Gao, Xingcheng Yao, and Danqi Chen. SimCSE: Simple contrastive learning of sentence embeddings. In *Proceedings of the 2021 Conference on Empirical Methods in Natural Language Processing*, pp. 6894–6910, Online and Punta Cana, Dominican Republic, November 2021. Association for Computational Linguistics. doi: 10.18653/v1/2021.emnlp-main.552. URL `https://aclanthology.org/2021.emnlp-main.552`.

Umang Gupta, Jwala Dhamala, Varun Kumar, Apurv Verma, Yada Pruksachatkun, Satyapriya Krishna, Rahul Gupta, Kai-Wei Chang, Greg Ver Steeg, and Aram Galstyan. Mitigating gender bias in distilled language models via counterfactual role reversal. In *Findings of the Association for Computational Linguistics: ACL 2022*, pp. 658–678, Dublin, Ireland, May 2022. Association for Computational Linguistics. doi: 10.18653/v1/2022.findings-acl.55. URL `https://aclanthology.org/2022.findings-acl.55`.

Kelvin Guu, Tatsunori B. Hashimoto, Yonatan Oren, and Percy Liang. Generating sentences by editing prototypes. *Transactions of the Association for Computational Linguistics*, 6:437–450, 2018. doi: 10.1162/tacl_a_00030. URL `https://aclanthology.org/Q18-1031`.

Dan Hendrycks, Collin Burns, Steven Basart, Andy Zou, Mantas Mazeika, Dawn Song, and Jacob Steinhardt. Measuring massive multitask language understanding. *arXiv preprint arXiv:2009.03300*, 2020.

Joel Hestness, Sharan Narang, Newsha Ardalani, Gregory Diamos, Heewoo Jun, Hassan Kianinejad, Md Patwary, Mostofa Ali, Yang Yang, and Yanqi Zhou. Deep learning scaling is predictable, empirically. *arXiv preprint arXiv:1712.00409*, 2017.

Ari Holtzman, Peter West, Vered Shwartz, Yejin Choi, and Luke Zettlemoyer. Surface form competition: Why the highest probability answer isn't always right. In *Proceedings of the 2021 Conference on Empirical Methods in Natural Language Processing*, pp. 7038–7051, Online

and Punta Cana, Dominican Republic, November 2021. Association for Computational Linguistics. doi: 10.18653/v1/2021.emnlp-main.564. URL https://aclanthology.org/2021.emnlp-main.564.

Minqing Hu and Bing Liu. Mining and summarizing customer reviews. In *Proceedings of the Tenth ACM SIGKDD International Conference on Knowledge Discovery and Data Mining*, KDD '04, pp. 168–177, New York, NY, USA, 2004. Association for Computing Machinery. ISBN 1581138881. doi: 10.1145/1014052.1014073. URL https://doi.org/10.1145/1014052.1014073.

Gautier Izacard, Patrick Lewis, Maria Lomeli, Lucas Hosseini, Fabio Petroni, Timo Schick, Jane Dwivedi-Yu, Armand Joulin, Sebastian Riedel, and Edouard Grave. Few-shot learning with retrieval augmented language models. *arXiv preprint arXiv:2208.03299*, 2022.

Zhengbao Jiang, Frank F. Xu, Jun Araki, and Graham Neubig. How can we know what language models know? *Transactions of the Association for Computational Linguistics*, 8:423–438, 2020. doi: 10.1162/tacl_a_00324. URL https://aclanthology.org/2020.tacl-1.28.

Zhengbao Jiang, Jun Araki, Haibo Ding, and Graham Neubig. How can we know when language models know? on the calibration of language models for question answering. *Transactions of the Association for Computational Linguistics*, 9:962–977, 2021. doi: 10.1162/tacl_a_00407. URL https://aclanthology.org/2021.tacl-1.57.

Lukasz Kaiser, Ofir Nachum, Aurko Roy, and Samy Bengio. Learning to remember rare events. In *International Conference on Learning Representations*, 2017. URL https://openreview.net/forum?id=SJTQLdqlg.

Urvashi Khandelwal, Omer Levy, Dan Jurafsky, Luke Zettlemoyer, and Mike Lewis. Generalization through memorization: Nearest neighbor language models. In *International Conference on Learning Representations*, 2020. URL https://openreview.net/forum?id=HklBjCEKvH.

Urvashi Khandelwal, Angela Fan, Dan Jurafsky, Luke Zettlemoyer, and Mike Lewis. Nearest neighbor machine translation. In *International Conference on Learning Representations*, 2021. URL https://openreview.net/forum?id=7wCBOfJ8hJM.

Takeshi Kojima, Shixiang Shane Gu, Machel Reid, Yutaka Matsuo, and Yusuke Iwasawa. Large language models are zero-shot reasoners. In *ICML 2022 Workshop on Knowledge Retrieval and Language Models*, 2022. URL https://openreview.net/forum?id=6p3AuaHAFiN.

Patrick Lewis, Ethan Perez, Aleksandra Piktus, Fabio Petroni, Vladimir Karpukhin, Naman Goyal, Heinrich Küttler, Mike Lewis, Wen-tau Yih, Tim Rocktäschel, Sebastian Riedel, and Douwe Kiela. Retrieval-augmented generation for knowledge-intensive nlp tasks. In H. Larochelle, M. Ranzato, R. Hadsell, M.F. Balcan, and H. Lin (eds.), *Advances in Neural Information Processing Systems*, volume 33, pp. 9459–9474. Curran Associates, Inc., 2020. URL https://proceedings.neurips.cc/paper/2020/file/6b493230205f780e1bc26945df7481e5-Paper.pdf.

Fangyu Liu, Yunlong Jiao, Jordan Massiah, Emine Yilmaz, and Serhii Havrylov. Trans-encoder: Unsupervised sentence-pair modelling through self- and mutual-distillations. In *International Conference on Learning Representations*, 2022a. URL https://openreview.net/forum?id=AmUhwTOHgm.

Jiachang Liu, Dinghan Shen, Yizhe Zhang, Bill Dolan, Lawrence Carin, and Weizhu Chen. What makes good in-context examples for GPT-3? In *Proceedings of Deep Learning Inside Out (DeeLIO 2022): The 3rd Workshop on Knowledge Extraction and Integration for Deep Learning Architectures*, pp. 100–114, Dublin, Ireland and Online, May 2022b. Association for Computational Linguistics. doi: 10.18653/v1/2022.deelio-1.10. URL https://aclanthology.org/2022.deelio-1.10.

Yao Lu, Max Bartolo, Alastair Moore, Sebastian Riedel, and Pontus Stenetorp. Fantastically ordered prompts and where to find them: Overcoming few-shot prompt order sensitivity. In *Proceedings of the 60th Annual Meeting of the Association for Computational Linguistics (Volume 1:*

*Long Papers)*, pp. 8086–8098, Dublin, Ireland, May 2022. Association for Computational Linguistics. doi: 10.18653/v1/2022.acl-long.556. URL `https://aclanthology.org/2022.acl-long.556`.

Sewon Min, Mike Lewis, Hannaneh Hajishirzi, and Luke Zettlemoyer. Noisy channel language model prompting for few-shot text classification. In *Proceedings of the 60th Annual Meeting of the Association for Computational Linguistics (Volume 1: Long Papers)*, pp. 5316–5330, Dublin, Ireland, May 2022a. Association for Computational Linguistics. doi: 10.18653/v1/2022.acl-long. 365. URL `https://aclanthology.org/2022.acl-long.365`.

Sewon Min, Mike Lewis, Luke Zettlemoyer, and Hannaneh Hajishirzi. MetaICL: Learning to learn in context. In *Proceedings of the 2022 Conference of the North American Chapter of the Association for Computational Linguistics: Human Language Technologies*, pp. 2791–2809, Seattle, United States, July 2022b. Association for Computational Linguistics. doi: 10.18653/v1/2022. naacl-main.201. URL `https://aclanthology.org/2022.naacl-main.201`.

Moin Nadeem, Anna Bethke, and Siva Reddy. StereoSet: Measuring stereotypical bias in pretrained language models. In *Proceedings of the 59th Annual Meeting of the Association for Computational Linguistics and the 11th International Joint Conference on Natural Language Processing (Volume 1: Long Papers)*, pp. 5356–5371, Online, August 2021. Association for Computational Linguistics. doi: 10.18653/v1/2021.acl-long.416. URL `https://aclanthology.org/2021.acl-long.416`.

Emin Orhan. A simple cache model for image recognition. In S. Bengio, H. Wallach, H. Larochelle, K. Grauman, N. Cesa-Bianchi, and R. Garnett (eds.), *Advances in Neural Information Processing Systems*, volume 31. Curran Associates, Inc., 2018. URL `https://proceedings.neurips.cc/paper/2018/file/6e0917469214d8fbd8c517dcdc6b8dcf-Paper.pdf`.

Bo Pang and Lillian Lee. A sentimental education: Sentiment analysis using subjectivity summarization based on minimum cuts. In *Proceedings of the 42nd Annual Meeting of the Association for Computational Linguistics (ACL-04)*, pp. 271–278, Barcelona, Spain, July 2004. doi: 10.3115/1218955.1218990. URL `https://aclanthology.org/P04-1035`.

Bo Pang and Lillian Lee. Seeing stars: Exploiting class relationships for sentiment categorization with respect to rating scales. In *Proceedings of the 43rd Annual Meeting of the Association for Computational Linguistics (ACL'05)*, pp. 115–124, Ann Arbor, Michigan, June 2005. Association for Computational Linguistics. doi: 10.3115/1219840.1219855. URL `https://aclanthology.org/P05-1015`.

Nicolas Papernot and Patrick McDaniel. Deep k-nearest neighbors: Towards confident, interpretable and robust deep learning. *arXiv preprint arXiv:1803.04765*, 2018.

Alec Radford, Jeffrey Wu, Rewon Child, David Luan, Dario Amodei, Ilya Sutskever, et al. Language models are unsupervised multitask learners. *OpenAI blog*, 1(8):9, 2019.

Jack W Rae, Sebastian Borgeaud, Trevor Cai, Katie Millican, Jordan Hoffmann, Francis Song, John Aslanides, Sarah Henderson, Roman Ring, Susannah Young, et al. Scaling language models: Methods, analysis & insights from training gopher. *arXiv preprint arXiv:2112.11446*, 2021.

Colin Raffel, Noam Shazeer, Adam Roberts, Katherine Lee, Sharan Narang, Michael Matena, Yanqi Zhou, Wei Li, and Peter J. Liu. Exploring the limits of transfer learning with a unified text-to-text transformer. *Journal of Machine Learning Research*, 21(140):1–67, 2020. URL `http://jmlr.org/papers/v21/20-074.html`.

Nils Reimers and Iryna Gurevych. Sentence-BERT: Sentence embeddings using Siamese BERT-networks. In *Proceedings of the 2019 Conference on Empirical Methods in Natural Language Processing and the 9th International Joint Conference on Natural Language Processing (EMNLP-IJCNLP)*, pp. 3982–3992, Hong Kong, China, November 2019. Association for Computational Linguistics. doi: 10.18653/v1/D19-1410. URL `https://aclanthology.org/D19-1410`.

Jonathan S. Rosenfeld, Amir Rosenfeld, Yonatan Belinkov, and Nir Shavit. A constructive prediction of the generalization error across scales. In *International Conference on Learning Representations*, 2020. URL `https://openreview.net/forum?id=ryenvpEKDr`.

Ohad Rubin, Jonathan Herzig, and Jonathan Berant. Learning to retrieve prompts for in-context learning. In *Proceedings of the 2022 Conference of the North American Chapter of the Association for Computational Linguistics: Human Language Technologies*, pp. 2655–2671, Seattle, United States, July 2022. Association for Computational Linguistics. doi: 10.18653/v1/2022. naacl-main.191. URL `https://aclanthology.org/2022.naacl-main.191`.

Victor Sanh, Albert Webson, Colin Raffel, Stephen Bach, Lintang Sutawika, Zaid Alyafeai, Antoine Chaffin, Arnaud Stiegler, Arun Raja, Manan Dey, M Saiful Bari, Canwen Xu, Urmish Thakker, Shanya Sharma Sharma, Eliza Szczechla, Taewoon Kim, Gunjan Chhablani, Nihal Nayak, Debajyoti Datta, Jonathan Chang, Mike Tian-Jian Jiang, Han Wang, Matteo Manica, Sheng Shen, Zheng Xin Yong, Harshit Pandey, Rachel Bawden, Thomas Wang, Trishala Neeraj, Jos Rozen, Abheesht Sharma, Andrea Santilli, Thibault Fevry, Jason Alan Fries, Ryan Teehan, Teven Le Scao, Stella Biderman, Leo Gao, Thomas Wolf, and Alexander M Rush. Multitask prompted training enables zero-shot task generalization. In *International Conference on Learning Representations*, 2022. URL `https://openreview.net/forum?id=9Vrb9D0WI4`.

Timo Schick and Hinrich Schütze. Exploiting cloze-questions for few-shot text classification and natural language inference. In *Proceedings of the 16th Conference of the European Chapter of the Association for Computational Linguistics: Main Volume*, pp. 255–269, Online, April 2021. Association for Computational Linguistics. doi: 10.18653/v1/2021.eacl-main.20. URL `https://aclanthology.org/2021.eacl-main.20`.

Weijia Shi, Julian Michael, Suchin Gururangan, and Luke Zettlemoyer. Nearest neighbor zero-shot inference. *arXiv preprint arXiv:2205.13792*, 2022.

Shaden Smith, Mostofa Patwary, Brandon Norick, Patrick LeGresley, Samyam Rajbhandari, Jared Casper, Zhun Liu, Shrimai Prabhumoye, George Zerveas, Vijay Korthikanti, et al. Using deepspeed and megatron to train megatron-turing nlg 530b, a large-scale generative language model. *arXiv preprint arXiv:2201.11990*, 2022.

Richard Socher, Alex Perelygin, Jean Wu, Jason Chuang, Christopher D. Manning, Andrew Ng, and Christopher Potts. Recursive deep models for semantic compositionality over a sentiment treebank. In *Proceedings of the 2013 Conference on Empirical Methods in Natural Language Processing*, pp. 1631–1642, Seattle, Washington, USA, October 2013. Association for Computational Linguistics. URL `https://aclanthology.org/D13-1170`.

Aarohi Srivastava, Abhinav Rastogi, Abhishek Rao, Abu Awal Md Shoeb, Abubakar Abid, Adam Fisch, Adam R Brown, Adam Santoro, Aditya Gupta, Adrià Garriga-Alonso, et al. Beyond the imitation game: Quantifying and extrapolating the capabilities of language models. *arXiv preprint arXiv:2206.04615*, 2022.

Trieu H Trinh and Quoc V Le. A simple method for commonsense reasoning. *arXiv preprint arXiv:1806.02847*, 2018.

Andrew Trotman, Antti Puurula, and Blake Burgess. Improvements to bm25 and language models examined. In *Proceedings of the 2014 Australasian Document Computing Symposium*, ADCS '14, pp. 58–65, New York, NY, USA, 2014. Association for Computing Machinery. ISBN 9781450330008. doi: 10.1145/2682862.2682863. URL `https://doi.org/10.1145/2682862.2682863`.

Laurens van der Maaten and Geoffrey Hinton. Visualizing data using t-sne. *Journal of Machine Learning Research*, 9(86):2579–2605, 2008. URL `http://jmlr.org/papers/v9/vandermaaten08a.html`.

Ellen M. Voorhees and Dawn M. Tice. Building a question answering test collection. In *Proceedings of the 23rd Annual International ACM SIGIR Conference on Research and Development in Information Retrieval*, SIGIR '00, pp. 200–207, New York, NY, USA, 2000. Association for Computing Machinery. ISBN 1581132263. doi: 10.1145/345508.345577. URL `https://doi.org/10.1145/345508.345577`.

Shuohang Wang, Yichong Xu, Yuwei Fang, Yang Liu, Siqi Sun, Ruochen Xu, Chenguang Zhu, and Michael Zeng. Training data is more valuable than you think: A simple and effective method by retrieving from training data. In *Proceedings of the 60th Annual Meeting of the Association for Computational Linguistics (Volume 1: Long Papers)*, pp. 3170–3179, Dublin, Ireland, May 2022. Association for Computational Linguistics. doi: 10.18653/v1/2022.acl-long.226. URL https://aclanthology.org/2022.acl-long.226.

Jason Wei, Maarten Bosma, Vincent Zhao, Kelvin Guu, Adams Wei Yu, Brian Lester, Nan Du, Andrew M. Dai, and Quoc V Le. Finetuned language models are zero-shot learners. In *International Conference on Learning Representations*, 2022a. URL https://openreview.net/forum?id=gEZrGCozdqR.

Jason Wei, Yi Tay, Rishi Bommasani, Colin Raffel, Barret Zoph, Sebastian Borgeaud, Dani Yogatama, Maarten Bosma, Denny Zhou, Donald Metzler, Ed H. Chi, Tatsunori Hashimoto, Oriol Vinyals, Percy Liang, Jeff Dean, and William Fedus. Emergent abilities of large language models. *Transactions on Machine Learning Research*, 2022b. URL https://openreview.net/forum?id=yzkSU5zdwD. Survey Certification.

Jason Wei, Xuezhi Wang, Dale Schuurmans, Maarten Bosma, Ed Chi, Quoc Le, and Denny Zhou. Chain of thought prompting elicits reasoning in large language models. *arXiv preprint arXiv:2201.11903*, 2022c.

Jason Weston, Emily Dinan, and Alexander Miller. Retrieve and refine: Improved sequence generation models for dialogue. In *Proceedings of the 2018 EMNLP Workshop SCAI: The 2nd International Workshop on Search-Oriented Conversational AI*, pp. 87–92, Brussels, Belgium, October 2018. Association for Computational Linguistics. doi: 10.18653/v1/W18-5713. URL https://www.aclweb.org/anthology/W18-5713.

Janyce Wiebe, Theresa Wilson, and Claire Cardie. Annotating expressions of opinions and emotions in language. *Language resources and evaluation*, 39(2):165–210, 2005.

Sang Michael Xie, Aditi Raghunathan, Percy Liang, and Tengyu Ma. An explanation of in-context learning as implicit bayesian inference. In *International Conference on Learning Representations*, 2022. URL https://openreview.net/forum?id=RdJVFCHjUMI.

Susan Zhang, Stephen Roller, Naman Goyal, Mikel Artetxe, Moya Chen, Shuohui Chen, Christopher Dewan, Mona Diab, Xian Li, Xi Victoria Lin, et al. Opt: Open pre-trained transformer language models. *arXiv preprint arXiv:2205.01068*, 2022.

Xiang Zhang, Junbo Zhao, and Yann LeCun. Character-level convolutional networks for text classification. In C. Cortes, N. Lawrence, D. Lee, M. Sugiyama, and R. Garnett (eds.), *Advances in Neural Information Processing Systems*, volume 28. Curran Associates, Inc., 2015. URL https://proceedings.neurips.cc/paper/2015/file/250cf8b51c773f3f8dc8b4be867a9a02-Paper.pdf.

Zihao Zhao, Eric Wallace, Shi Feng, Dan Klein, and Sameer Singh. Calibrate before use: Improving few-shot performance of language models. In Marina Meila and Tong Zhang (eds.), *Proceedings of the 38th International Conference on Machine Learning*, volume 139 of *Proceedings of Machine Learning Research*, pp. 12697–12706. PMLR, 18–24 Jul 2021. URL https://proceedings.mlr.press/v139/zhao21c.html.

## A  DATASET STATISTICS: MAX SHOT IN CONTEXT

We provide detailed statistics about the number of maximum shots in Table 5, i.e., $M_T$ for each task, corresponding to Table 1 in the main manuscript.

| | | SST2 | SUBJ | MPQA | AGNews | CB | CR | DBPedia | MR | RTE | TREC |
|---|---|---|---|---|---|---|---|---|---|---|---|
| Num. of Classes | | 2 | 2 | 2 | 4 | 3 | 2 | 14 | 2 | 2 | 6 |
| Average Length of Templates | | 19.1 | 34.9 | 10.4 | 59.5 | 90.8 | 29.0 | 71.6 | 32.7 | 79.8 | 17.6 |
| 1024 Toks | Max Shots (TP) | 20 (2%) | 12 (1%) | 39 (0%) | 3 (0%) | 2 (0%) | 14 (4%) | 1 (77%) | 14 (4%) | 4 (0%) | 8 (1%) |
| | $M_T$ | 16 | 8 | 32 | 2 | 2 | 8 | 1 | 8 | 4 | 8 |
| 2048 Toks | Max Shots (TP) | 44 (5%) | 25 (1%) | 81 (3%) | 7 (1%) | 6 (2%) | 28 (4%) | 1 (0%) | 27 (2%) | 10 (4%) | 17 (5%) |
| | $M_T$ | 32 | 16 | 32 | 4 | 4 | 16 | 1 | 16 | 8 | 16 |

Table 5: Dataset statistics and the maximum shots (per class) that a context of 1024 tokens or 2048 tokens can allow. We provide maximum shots under 5% Truncation Probability (TP) restriction as well as the actual $M_T$ taken in this paper, which is set from $\{1, 2, 4, 8, 16, 32\}$ for simplicity.

## B  MORE ANALYSES

### B.1  DISTANCE MEASUREMENT

We investigate euclidean distance as an alternative distance measurement, which has also been explored in Khandelwal et al. (2020). We take the contextual representation $h$ of LLM, and denotes their distance as $D_{L2}(h_{\text{test}}, h_i)$. Table 6 shows that both methods are effective but $D_{KL}$ (Equation 5) based on LM distribution $p$ is a superior measurement. Actually, $p$ is a projection of $h$ through the word embedding, we think this procedure exploits the well-structured word embeddings of LLM to provide more disentangled representations, thus can better serves as the anchor space.

| Measurements | SST2 | SUBJ | MPQA | AGNews | CB | CR | DBPedia | MR | RTE | TREC | AVG |
|---|---|---|---|---|---|---|---|---|---|---|---|
| In-Context Learning | $81.3_{\pm5.4}$ | $64.1_{\pm11.3}$ | $75.2_{\pm8.8}$ | $72.7_{\pm18.5}$ | $\mathbf{60.7}_{\pm2.8}$ | $66.2_{\pm16.7}$ | $83.5_{\pm3.8}$ | $72.2_{\pm13.9}$ | $53.0_{\pm1.7}$ | $54.2_{\pm4.9}$ | 68.31 |
| Contextual Repr + $D_{L2}$ | $84.1_{\pm8.5}$ | $73.0_{\pm8.4}$ | $75.2_{\pm8.8}$ | $82.0_{\pm0.8}$ | $60.4_{\pm8.1}$ | $78.0_{\pm11.7}$ | $\mathbf{95.1}_{\pm1.0}$ | $82.6_{\pm5.0}$ | $53.3_{\pm2.4}$ | $\mathbf{70.3}_{\pm4.4}$ | 75.40 (+7.09) |
| LM Distribution + $D_{KL}$ | $\mathbf{87.7}_{\pm3.5}$ | $\mathbf{77.0}_{\pm3.5}$ | $75.2_{\pm8.8}$ | $\mathbf{86.2}_{\pm1.8}$ | $58.9_{\pm2.2}$ | $\mathbf{88.2}_{\pm3.5}$ | $94.1_{\pm2.3}$ | $\mathbf{83.9}_{\pm2.4}$ | $\mathbf{53.6}_{\pm3.0}$ | $64.8_{\pm4.2}$ | $\mathbf{76.97}$ (+8.66) |

Table 6: Comparison for alternative distance measurement.

### B.2  RELIANCE ON PRIOR KNOWLEDGE

**Conclusion**  We further explore the robustness of $k$NN Prompting regarding the reliance of prior knowledge on target distribution. We show that among the investigated baselines, ICL and ContextualCalibration (Zhao et al., 2021) are greatly impacted by prior knowledge of test distribution, while $k$NN Prompting and NoisyChannel (Min et al., 2022a) are much more robust.

**Experimental Setting**  We investigate various combinations of prior distribution by controlling the imbalance ratio $\lambda_{train}$ and $\lambda_{test}$. For every setting, we include 5 binary classification tasks (SST2, MPQA, SUBJ, MR, CR) and run with 10 random seeds, we report the average score of these results. Specifically, $\lambda_{train} = 0.125$ means one category (positive) accounts for 12.5% of the entire train set, and $\lambda_{train} = 1.5$ corresponds to the balanced setting. We investigate both $\lambda_{train} < 0.5$ and $\lambda_{test} < 0.5$, which results in three different settings.

ContextualCalibration explicitly include a prior distribution, at such imbalanced scenario, one can either use the default assumption ($pos : neg = 1 : 1$, as designed in the original paper) or use the observed assumption from train set ($pos : neg = \lambda_{train}/(1 - \lambda_{train})$). We respectively refer to them as **Balanced Prior** and **Trainset Prior**. Note that test distribution is unaccessible so we can not use it. Other methods (ICL, NoisyChannel and $k$NN Prompting) do not *technically* incorporate any prior knowledge. So they are not concerned with this investigation dimension.

**Analyses**  The results are reported in Table 7. For ICL, LLM naturally suffers from the bias learned in pretraining stage, thus is vulnerable to any different prior distributions. The performance greatly degrades in $Setting\ B$ (-22.06) and $Setting\ C$ (-22.17).

For ContextualCalibration, technically, it necessarily requires a prior distribution to rectify the LLM predicted label word probabilities. If the prior knowledge does not match the (unaccessible) test distribution, its performance will be greatly degraded. Specifically, if Balanced Prior is consistent with test distribution, the method performs well ($Setting\ A$, -1.77), otherwise, the performance degrades ($Setting\ B$, -20.24 and $Setting\ C$, -12.65). Similarly, if Trainset Prior Assumption is consistent with test distribution, the method performs well ($Setting\ C$, +8.65), otherwise, the performance degrades ($Setting\ B$, -20.24 and $Setting\ A$, -18.92).

For NoisyChannel, its performance is rather robust (-2.60/-0.63/-7.87 respectively in $Setting\ A, B, C$). By re-formulating ICL into computing conditional probability of the input given the output, it is indeed an effective way to calibrate the task prediction.

For the proposed $k$NN Prompting, technically, it does **not** incorporate any prior knowledge of train or test distribution. Both the construction of datastore and the retrieving then predicting procedure do not vary w.r.t. different prior knowledge of distribution. The proposed method can robustly adapt to all imbalanced settings, including imbalanced trainset, testset and both. There is basically no performance degradation (-1.3/-2.6/+0.05 respectively in $Setting\ A, B, C$, where -1.3/-2.6 can be considered within ordinary fluctuation).

| Methods | 0.125 | 0.25 | 0.375 | 0.5 (Balanced) | AVG | MaxDrop |
|---|---|---|---|---|---|---|
| *Setting A.* $\lambda_{train} < 0.5, \lambda_{test} = 0.5$ | | | | | | |
| ICL | 62.4 | 64.2 | 65.7 | 68.2 | 65.15 | -5.85 |
| **ContextualCalibration w/ Balanced Prior** | 74.0 | 69.7 | 68.5 | 70.3 | 70.64 | -1.77 |
| **ContextualCalibration w/ Trainset Prior** | 51.4 | 71.5 | 80.2 | 70.3 | 68.36 | -18.92 |
| **NoisyChannel** | 70.0 | 70.9 | 71.9 | 72.6 | 71.33 | -2.60 |
| *k*NN Prompting | **79.0** | **80.1** | **80.5** | **80.3** | **79.98** | **-1.29** |
| *Setting B.* $\lambda_{train} = 0.5, \lambda_{test} < 0.5$ | | | | | | |
| ICL | 45.9 | 53.4 | 60.9 | 67.9 | 57.04 | -22.06 |
| **ContextualCalibration w/ Balanced Prior** | 50.0 | 57.2 | 64.2 | 70.2 | 60.42 | -20.24 |
| **ContextualCalibration w/ Trainset Prior** | 50.0 | 57.2 | 64.2 | 70.2 | 60.42 | -20.24 |
| **NoisyChannel** | 72.1 | 72.2 | 71.8 | 72.4 | 72.14 | **-0.63** |
| *k*NN Prompting | **77.1** | **78.0** | **78.7** | **79.7** | **78.38** | -2.62 |
| *Setting C.* $\lambda_{train} = \lambda_{test} < 0.5$ | | | | | | |
| ICL | 45.8 | 50.0 | 58.6 | 67.9 | 55.57 | -22.17 |
| **ContextualCalibration w/ Balanced Prior** | 63.0 | 57.6 | 61.7 | 70.2 | 63.13 | -12.65 |
| **ContextualCalibration w/ Trainset Prior** | **87.8** | **84.5** | 78.8 | 70.2 | **80.33** | **+8.56** |
| **NoisyChannel** | 64.6 | 68.6 | 70.8 | 72.4 | 69.10 | -7.87 |
| *k*NN Prompting | 79.8 | 80.7 | **80.3** | **79.7** | 80.12 | **+0.05** |

Table 7: Reliance on prior knowledge. All reported results are averaged across 5 datasets, and we further report average performance across all imbalance ratios. $\lambda_{train/test}$ denotes the subsampled ratio. MaxDrop measures the performance degradation compared to ordinary balanced setting ($\lambda_* = 0.5$), where the best is bolded and no drop is highlighted in **pink**.

### B.3 EMPIRICAL CHOICE OF $k$

In Figure 10 we search for different choices of $k$ on MPQA, and found that it is generally a rather robust choice within the wide range $[3, |\mathcal{A}|/2 - 1]$.

### B.4 ROBUSTNESS UNDER IMBALANCED SCENARIO

We further test the robustness of $k$NN Prompting under imbalanced label scenario. Take binary classification like SST2 as example, we simulate imbalance ratio by controlling one of the category proportionally to {0.5, 0.25, 0.125, 0.0625, 0.03125, 0.015625} of the entire training set, where 0.5 corresponds to the ordinary balanced scenario. We keep the test set intact, which results in a challenging out-of-distribution (OOD) setting. Results in Figure 11 reveal the vulnerability of the proposed method. Under imbalanced setting, $k$NN Prompting is overwhelmed by the large quantity of anchors from the majority class where it is simply far more easier to find closer neighbors.

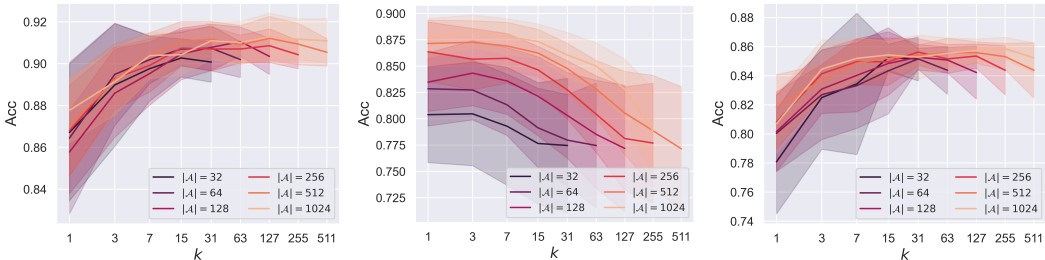

Figure 10: Empirical choices of $k$. Conducted on SST2, SUBJ and MPQA respectively (left to right).

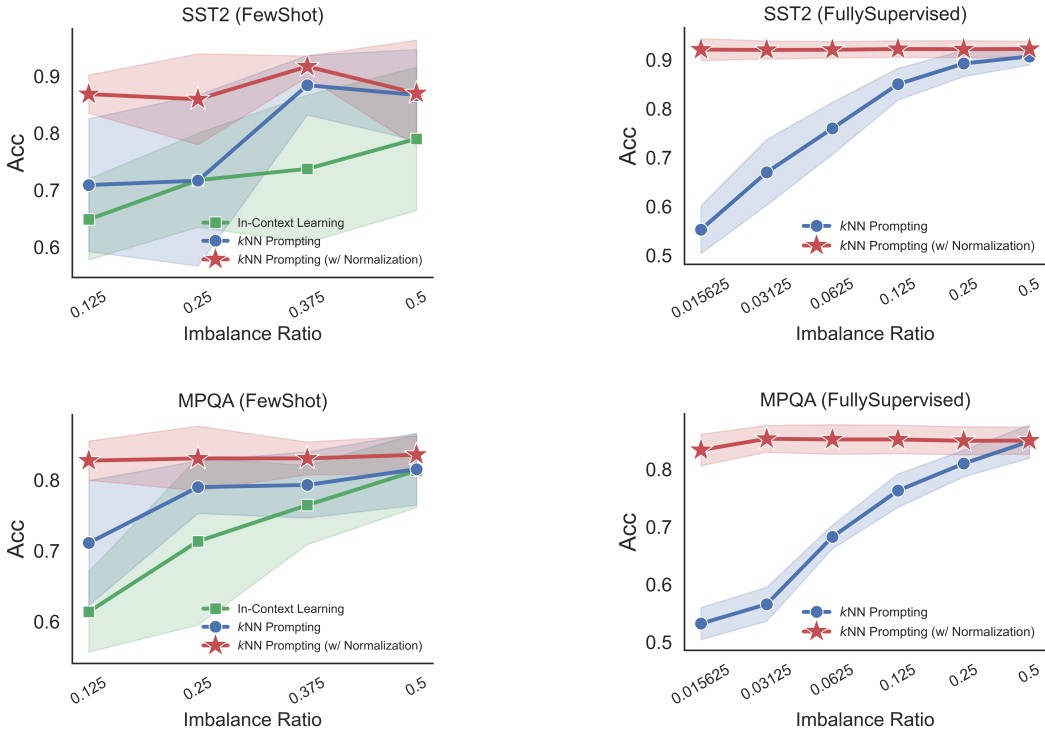

Figure 11: Robustness under imbalanced scenario. Left: few-shot scenario (32). Right: fully-supervised scenario (1024).

To address such performance degradation under challenging imbalanced scenario, we propose a simple normalization trick: we average the anchor representations to produce one centered anchor for each class, the resulting anchor is thus more representative and also avoids quantity distraction. Such an adaptation works surprisingly well with no loss of performance even under ordinary balanced setting.

## C  COMPLETE RESULTS

### C.1  DATA SCALING CURVE UNDER FEW-SHOT SCENARIO

We provide scaling curve w.r.t. each specific dataset in Figure 12, corresponding to Table 2 and Figure 4 in the main manuscript.

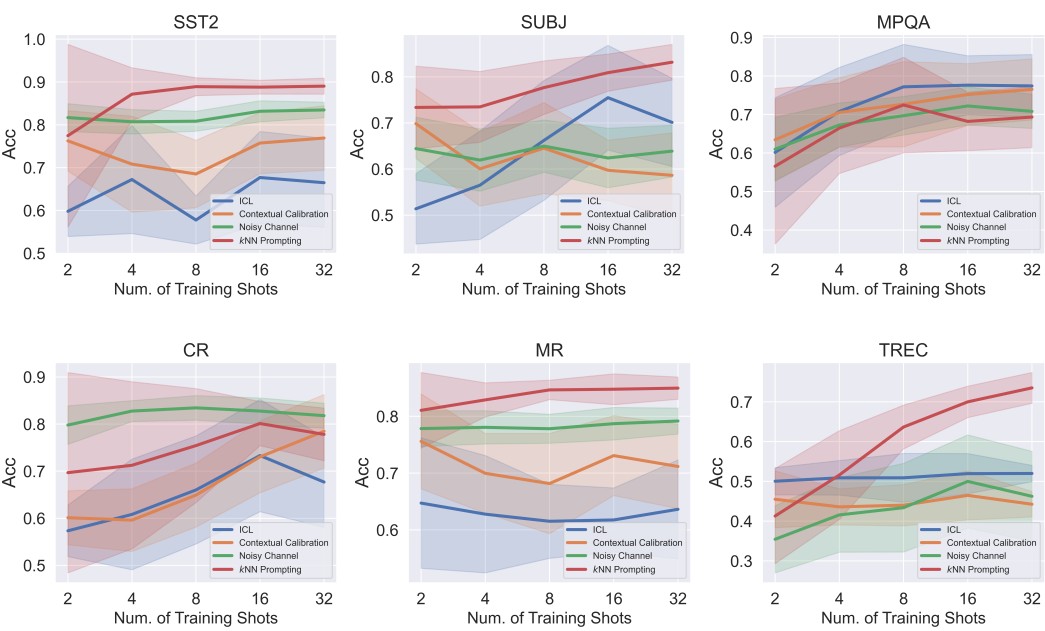

Figure 12: Scaling curve under few shot setting for each specific dataset. The baselines are strictly comparable under $m \le 8$, some baselines might be truncated when $m \ge 16$.

| Models & Methods | SST2 | SUBJ | MPQA | AGNews | CB | CR | DBPedia | MR | RTE | TREC | AVG |
|---|---|---|---|---|---|---|---|---|---|---|---|
| **In-Context Learning** | $63.4_{\pm7.3}$ | $58.9_{\pm8.7}$ | $70.5_{\pm5.2}$ | $61.7_{\pm15.4}$ | $45.0_{\pm9.1}$ | $83.3_{\pm13.7}$ | $59.9_{\pm11.5}$ | $77.0_{\pm15.7}$ | $53.6_{\pm3.1}$ | $54.4_{\pm1.7}$ | 62.77 |
| $m=32$ | $82.0_{\pm6.9}$ | $82.8_{\pm1.2}$ | $70.5_{\pm5.2}$ | $82.2_{\pm0.9}$ | $56.8_{\pm7.3}$ | $87.5_{\pm3.0}$ | $93.7_{\pm1.1}$ | $83.0_{\pm2.5}$ | $53.6_{\pm4.0}$ | $71.5_{\pm6.2}$ | 76.35 |
| $m=64$ | $84.2_{\pm5.4}$ | $84.6_{\pm0.7}$ | $81.2_{\pm3.0}$ | $83.3_{\pm1.9}$ | $59.3_{\pm6.0}$ | $90.3_{\pm1.4}$ | $95.2_{\pm0.8}$ | $82.2_{\pm2.9}$ | $53.8_{\pm1.7}$ | $72.4_{\pm4.1}$ | 78.66 |
| 0.8B $\quad m=128$ | $84.5_{\pm5.3}$ | $85.8_{\pm1.6}$ | $83.1_{\pm0.8}$ | $84.5_{\pm1.3}$ | $62.1_{\pm3.4}$ | $89.7_{\pm0.6}$ | $95.8_{\pm0.5}$ | $84.0_{\pm1.8}$ | $53.6_{\pm3.2}$ | $74.2_{\pm4.4}$ | 79.73 |
| $m=256$ | $85.8_{\pm4.0}$ | $86.0_{\pm2.5}$ | $83.2_{\pm1.3}$ | $86.5_{\pm1.3}$ | $62.1_{\pm3.4}$ | $89.5_{\pm0.4}$ | $96.1_{\pm0.9}$ | $82.8_{\pm0.9}$ | $53.5_{\pm3.3}$ | $81.0_{\pm2.5}$ | 80.66 |
| $m=512$ | $85.5_{\pm4.2}$ | $86.7_{\pm1.5}$ | $83.0_{\pm2.4}$ | $86.3_{\pm0.7}$ | $62.1_{\pm3.4}$ | $89.1_{\pm1.0}$ | $96.4_{\pm0.8}$ | $83.5_{\pm4.1}$ | $52.8_{\pm3.4}$ | $82.4_{\pm2.9}$ | 80.80 |
| $m=1024$ | $85.5_{\pm4.5}$ | $87.6_{\pm1.4}$ | $84.8_{\pm1.0}$ | $87.6_{\pm0.6}$ | $62.1_{\pm3.4}$ | $87.5_{\pm1.0}$ | $96.7_{\pm0.7}$ | $83.6_{\pm2.3}$ | $54.1_{\pm2.6}$ | $86.7_{\pm1.5}$ | 81.61 |
| **In-Context Learning** | $81.3_{\pm5.4}$ | $64.1_{\pm11.3}$ | $75.2_{\pm8.8}$ | $72.7_{\pm18.5}$ | $60.7_{\pm2.8}$ | $66.2_{\pm16.7}$ | $83.5_{\pm3.8}$ | $72.2_{\pm13.9}$ | $53.0_{\pm1.7}$ | $54.2_{\pm4.9}$ | 68.31 |
| $m=32$ | $87.7_{\pm3.5}$ | $77.0_{\pm3.5}$ | $75.2_{\pm8.8}$ | $86.2_{\pm1.8}$ | $58.9_{\pm2.2}$ | $88.2_{\pm3.5}$ | $94.1_{\pm2.3}$ | $83.9_{\pm2.4}$ | $53.6_{\pm3.0}$ | $64.8_{\pm4.2}$ | 76.97 |
| $m=64$ | $87.2_{\pm2.8}$ | $82.5_{\pm1.2}$ | $83.0_{\pm4.0}$ | $86.2_{\pm1.1}$ | $62.9_{\pm6.1}$ | $89.1_{\pm1.6}$ | $95.5_{\pm1.4}$ | $84.3_{\pm1.9}$ | $49.1_{\pm3.2}$ | $68.9_{\pm2.9}$ | 78.86 |
| 1.5B $\quad m=128$ | $86.3_{\pm2.9}$ | $83.8_{\pm2.1}$ | $82.3_{\pm1.5}$ | $87.2_{\pm0.4}$ | $64.6_{\pm7.9}$ | $88.9_{\pm2.4}$ | $96.5_{\pm0.7}$ | $86.4_{\pm0.8}$ | $51.1_{\pm1.7}$ | $74.0_{\pm2.9}$ | 80.12 |
| $m=256$ | $89.1_{\pm1.6}$ | $85.0_{\pm2.2}$ | $83.4_{\pm1.5}$ | $87.5_{\pm1.7}$ | $64.6_{\pm7.9}$ | $89.3_{\pm0.9}$ | $97.3_{\pm0.3}$ | $85.6_{\pm2.0}$ | $52.7_{\pm3.3}$ | $78.1_{\pm3.3}$ | 81.27 |
| $m=512$ | $89.0_{\pm1.5}$ | $87.6_{\pm1.6}$ | $83.7_{\pm1.2}$ | $88.4_{\pm1.5}$ | $64.6_{\pm7.9}$ | $90.1_{\pm1.0}$ | $97.6_{\pm0.5}$ | $85.1_{\pm1.6}$ | $52.9_{\pm1.8}$ | $80.5_{\pm1.8}$ | 81.94 |
| $m=1024$ | $88.2_{\pm1.0}$ | $88.4_{\pm1.8}$ | $84.1_{\pm1.2}$ | $89.1_{\pm1.2}$ | $64.6_{\pm7.9}$ | $86.7_{\pm1.3}$ | $97.8_{\pm0.6}$ | $84.4_{\pm1.5}$ | $53.3_{\pm4.2}$ | $83.0_{\pm1.4}$ | 81.96 |
| **In-Context Learning** | $89.9_{\pm4.5}$ | $77.7_{\pm5.8}$ | $84.5_{\pm2.1}$ | $78.8_{\pm4.0}$ | $51.1_{\pm6.1}$ | $92.6_{\pm0.6}$ | $88.8_{\pm1.3}$ | $92.5_{\pm1.0}$ | $52.4_{\pm4.2}$ | $64.6_{\pm3.0}$ | 77.29 |
| $m=32$ | $89.9_{\pm4.5}$ | $82.3_{\pm4.4}$ | $84.5_{\pm2.1}$ | $84.4_{\pm1.2}$ | $50.0_{\pm5.4}$ | $89.9_{\pm3.2}$ | $97.0_{\pm0.7}$ | $91.2_{\pm2.6}$ | $52.8_{\pm1.9}$ | $75.1_{\pm5.5}$ | 79.72 |
| $m=64$ | $91.4_{\pm2.1}$ | $84.9_{\pm3.7}$ | $80.8_{\pm6.0}$ | $86.2_{\pm1.7}$ | $57.1_{\pm3.3}$ | $91.0_{\pm1.7}$ | $97.7_{\pm0.8}$ | $91.5_{\pm1.9}$ | $51.6_{\pm3.4}$ | $79.1_{\pm2.8}$ | 81.14 |
| 2.7B $\quad m=128$ | $93.4_{\pm1.3}$ | $87.5_{\pm2.0}$ | $83.3_{\pm3.9}$ | $86.7_{\pm2.4}$ | $57.1_{\pm3.8}$ | $90.2_{\pm2.0}$ | $98.6_{\pm0.6}$ | $91.4_{\pm1.5}$ | $53.4_{\pm5.6}$ | $80.8_{\pm2.0}$ | 82.25 |
| $m=256$ | $92.9_{\pm2.0}$ | $87.7_{\pm1.7}$ | $83.3_{\pm2.4}$ | $86.4_{\pm1.3}$ | $57.1_{\pm3.8}$ | $91.6_{\pm1.5}$ | $99.0_{\pm0.2}$ | $90.5_{\pm1.7}$ | $53.5_{\pm3.2}$ | $81.2_{\pm2.5}$ | 82.32 |
| $m=512$ | $91.9_{\pm1.0}$ | $89.4_{\pm1.0}$ | $83.8_{\pm2.3}$ | $88.1_{\pm2.1}$ | $57.1_{\pm3.8}$ | $91.3_{\pm1.2}$ | $99.1_{\pm0.4}$ | $91.2_{\pm1.4}$ | $54.3_{\pm1.6}$ | $83.1_{\pm4.0}$ | 82.93 |
| $m=1024$ | $91.7_{\pm2.3}$ | $90.5_{\pm1.5}$ | $83.7_{\pm2.2}$ | $89.1_{\pm1.2}$ | $57.1_{\pm3.8}$ | $89.8_{\pm1.0}$ | $99.1_{\pm0.3}$ | $90.9_{\pm1.2}$ | $57.7_{\pm3.0}$ | $86.7_{\pm2.3}$ | 83.64 |
| **In-Context Learning** | $92.7_{\pm1.8}$ | $81.6_{\pm4.6}$ | $86.2_{\pm1.9}$ | $68.8_{\pm8.2}$ | $52.5_{\pm9.7}$ | $92.0_{\pm2.6}$ | $89.8_{\pm0.9}$ | $91.6_{\pm1.1}$ | $55.0_{\pm1.6}$ | $62.0_{\pm7.3}$ | 77.19 |
| $m=32$ | $92.7_{\pm1.8}$ | $85.5_{\pm2.9}$ | $86.2_{\pm1.9}$ | $83.0_{\pm1.5}$ | $65.0_{\pm1.6}$ | $91.9_{\pm3.2}$ | $97.4_{\pm0.6}$ | $91.8_{\pm0.7}$ | $55.2_{\pm3.2}$ | $71.2_{\pm5.2}$ | 81.90 |
| $m=64$ | $92.7_{\pm0.8}$ | $87.2_{\pm2.0}$ | $85.2_{\pm3.9}$ | $85.1_{\pm1.4}$ | $66.4_{\pm6.6}$ | $90.5_{\pm2.8}$ | $98.1_{\pm0.5}$ | $91.1_{\pm1.4}$ | $51.6_{\pm4.4}$ | $72.3_{\pm1.3}$ | 82.02 |
| 6B $\quad m=128$ | $92.2_{\pm1.2}$ | $87.4_{\pm1.7}$ | $85.2_{\pm2.5}$ | $87.4_{\pm1.6}$ | $63.6_{\pm4.1}$ | $90.6_{\pm2.2}$ | $98.5_{\pm0.4}$ | $91.0_{\pm1.5}$ | $55.6_{\pm2.0}$ | $75.8_{\pm3.1}$ | 82.73 |
| $m=256$ | $92.9_{\pm0.8}$ | $90.5_{\pm1.2}$ | $84.9_{\pm0.9}$ | $86.4_{\pm2.0}$ | $63.6_{\pm4.1}$ | $90.7_{\pm0.7}$ | $99.0_{\pm0.2}$ | $91.4_{\pm1.0}$ | $56.2_{\pm2.4}$ | $76.6_{\pm1.5}$ | 83.22 |
| $m=512$ | $93.5_{\pm1.3}$ | $90.8_{\pm0.9}$ | $85.5_{\pm1.7}$ | $88.6_{\pm0.8}$ | $63.6_{\pm4.1}$ | $90.4_{\pm1.4}$ | $99.0_{\pm0.2}$ | $90.3_{\pm1.0}$ | $59.4_{\pm2.0}$ | $81.6_{\pm1.4}$ | 84.27 |
| $m=1024$ | $93.0_{\pm1.1}$ | $90.6_{\pm1.6}$ | $85.5_{\pm1.4}$ | $89.5_{\pm1.7}$ | $63.6_{\pm4.1}$ | $88.3_{\pm1.6}$ | $98.9_{\pm0.5}$ | $90.4_{\pm1.5}$ | $57.0_{\pm2.7}$ | $81.5_{\pm0.9}$ | 83.82 |
| **In-Context Learning** | $89.0_{\pm4.3}$ | $91.3_{\pm2.4}$ | $78.4_{\pm7.2}$ | $78.1_{\pm5.6}$ | $53.2_{\pm4.4}$ | $93.4_{\pm1.1}$ | $92.2_{\pm2.4}$ | $89.9_{\pm2.2}$ | $55.8_{\pm3.0}$ | $55.5_{\pm6.1}$ | 77.68 |
| $m=32$ | $89.0_{\pm4.3}$ | $91.1_{\pm1.1}$ | $78.4_{\pm7.2}$ | $84.5_{\pm2.2}$ | $77.9_{\pm4.1}$ | $89.5_{\pm4.6}$ | $97.3_{\pm0.7}$ | $91.1_{\pm1.4}$ | $56.8_{\pm3.2}$ | $71.4_{\pm4.1}$ | 82.69 |
| $m=64$ | $94.8_{\pm0.8}$ | $90.8_{\pm2.1}$ | $86.3_{\pm2.7}$ | $86.0_{\pm1.1}$ | $77.5_{\pm5.1}$ | $91.8_{\pm1.1}$ | $98.1_{\pm0.7}$ | $91.1_{\pm2.1}$ | $57.7_{\pm4.1}$ | $70.5_{\pm2.7}$ | 84.47 |
| 13B $\quad m=128$ | $94.8_{\pm0.6}$ | $90.1_{\pm1.5}$ | $86.2_{\pm3.8}$ | $87.4_{\pm1.7}$ | $78.9_{\pm4.8}$ | $89.6_{\pm1.1}$ | $98.9_{\pm0.5}$ | $92.0_{\pm0.5}$ | $58.2_{\pm4.4}$ | $73.1_{\pm2.9}$ | 84.93 |
| $m=256$ | $94.5_{\pm0.9}$ | $90.5_{\pm1.6}$ | $85.3_{\pm0.7}$ | $87.2_{\pm1.8}$ | $78.9_{\pm4.8}$ | $91.2_{\pm1.9}$ | $99.5_{\pm0.2}$ | $92.0_{\pm1.4}$ | $59.3_{\pm1.5}$ | $74.8_{\pm4.1}$ | 85.31 |
| $m=512$ | $94.5_{\pm0.8}$ | $91.2_{\pm2.1}$ | $86.3_{\pm0.9}$ | $86.9_{\pm1.5}$ | $78.9_{\pm4.8}$ | $90.5_{\pm1.3}$ | $99.3_{\pm0.3}$ | $92.7_{\pm0.3}$ | $64.1_{\pm4.8}$ | $75.9_{\pm4.1}$ | 86.03 |
| $m=1024$ | $94.5_{\pm0.4}$ | $91.6_{\pm2.4}$ | $87.0_{\pm2.1}$ | $88.5_{\pm1.0}$ | $78.9_{\pm4.8}$ | $88.1_{\pm1.3}$ | $99.3_{\pm0.2}$ | $91.6_{\pm0.3}$ | $63.0_{\pm3.6}$ | $76.1_{\pm6.2}$ | 85.87 |
| **In-Context Learning** | $90.8_{\pm4.1}$ | $83.5_{\pm8.7}$ | $80.7_{\pm1.9}$ | $74.8_{\pm4.6}$ | $64.6_{\pm8.3}$ | $87.7_{\pm3.9}$ | $93.3_{\pm0.4}$ | $93.4_{\pm1.0}$ | $61.6_{\pm2.7}$ | $71.7_{\pm2.7}$ | 80.21 |
| $m=32$ | $90.8_{\pm4.1}$ | $91.3_{\pm0.6}$ | $80.7_{\pm1.9}$ | $84.1_{\pm1.2}$ | $70.0_{\pm9.9}$ | $89.0_{\pm3.5}$ | $98.3_{\pm0.2}$ | $94.5_{\pm1.1}$ | $59.5_{\pm3.6}$ | $77.6_{\pm2.3}$ | 83.56 |
| $m=64$ | $93.6_{\pm1.6}$ | $91.9_{\pm1.4}$ | $85.5_{\pm1.8}$ | $85.3_{\pm2.1}$ | $69.3_{\pm7.6}$ | $91.6_{\pm1.6}$ | $98.3_{\pm0.2}$ | $94.1_{\pm0.6}$ | $60.2_{\pm6.0}$ | $81.0_{\pm2.9}$ | 85.08 |
| 30B $\quad m=128$ | $94.3_{\pm0.9}$ | $92.7_{\pm1.7}$ | $84.5_{\pm0.9}$ | $87.1_{\pm1.4}$ | $70.7_{\pm8.2}$ | $91.0_{\pm1.2}$ | $98.8_{\pm0.5}$ | $93.1_{\pm1.5}$ | $61.7_{\pm4.8}$ | $79.8_{\pm2.0}$ | 85.38 |
| $m=256$ | $94.4_{\pm0.4}$ | $93.8_{\pm1.1}$ | $84.1_{\pm1.1}$ | $87.6_{\pm0.9}$ | $70.7_{\pm8.2}$ | $90.3_{\pm1.0}$ | $99.1_{\pm0.3}$ | $93.7_{\pm0.6}$ | $62.1_{\pm2.5}$ | $82.1_{\pm1.9}$ | 85.80 |
| $m=512$ | $94.1_{\pm0.3}$ | $94.1_{\pm1.4}$ | $85.6_{\pm1.5}$ | $87.9_{\pm1.9}$ | $70.7_{\pm8.2}$ | $89.8_{\pm1.5}$ | $99.1_{\pm0.3}$ | $93.2_{\pm1.1}$ | $61.8_{\pm2.6}$ | $83.8_{\pm1.8}$ | 86.01 |
| $m=1024$ | $94.1_{\pm0.6}$ | $93.9_{\pm0.8}$ | $85.4_{\pm0.7}$ | $87.9_{\pm2.3}$ | $70.7_{\pm8.2}$ | $88.4_{\pm1.1}$ | $99.1_{\pm0.2}$ | $93.4_{\pm1.6}$ | $61.0_{\pm4.0}$ | $86.2_{\pm1.5}$ | 86.02 |

Table 8: Full results for data scaling, corresponds to Figure 5. Some ICL results are reused[7].

## C.2 Data Scaling Results Under Fully Supervised Scenario

We provide comprehensive results of $k$NN Prompting across data scales and LLM scales in Table 8, corresponding to Figure 5 in the main manuscript.

## C.3 Full Results of Comparison to PromptCompose

We provide the full results of comparison between $k$NN Prompting and PromptCompose in Table 9, corresponding to Figure 6 in the main manuscript. The employed sentence encoder can be found at `https://huggingface.co/models`[8][9][10].

| Setting & Methods | | SST2 | SUBJ | MPQA | AGNews | CB | CR | DBPedia | MR | RTE | TREC | AVG |
|---|---|---|---|---|---|---|---|---|---|---|---|---|
| $m = M_T$ | **In-Context Learning** | $81.3_{\pm5.4}$ | $64.1_{\pm11.3}$ | $75.2_{\pm8.8}$ | $72.7_{\pm18.5}$ | $60.7_{\pm2.8}$ | $66.2_{\pm16.7}$ | $83.5_{\pm3.8}$ | $72.2_{\pm13.9}$ | $53.0_{\pm1.7}$ | $54.2_{\pm4.9}$ | 68.31 |
| | **BM25** | $68.4_{\pm3.7}$ | $63.6_{\pm3.0}$ | $75.2_{\pm8.8}$ | $69.0_{\pm3.6}$ | $65.0_{\pm6.5}$ | $55.2_{\pm1.0}$ | $80.7_{\pm1.3}$ | $59.4_{\pm1.4}$ | $53.0_{\pm2.3}$ | $65.5_{\pm2.8}$ | 65.50 |
| | **SBERT** | $71.0_{\pm4.9}$ | $67.5_{\pm1.6}$ | $75.2_{\pm8.8}$ | $82.3_{\pm2.1}$ | $62.9_{\pm2.9}$ | $57.8_{\pm1.9}$ | $83.8_{\pm1.2}$ | $57.7_{\pm4.3}$ | $51.2_{\pm3.1}$ | $58.4_{\pm6.7}$ | 66.78 |
| $m = 32$ | **SimCSE** | $68.1_{\pm4.5}$ | $69.8_{\pm3.1}$ | $75.2_{\pm8.8}$ | $80.7_{\pm2.9}$ | $66.4_{\pm2.3}$ | $55.9_{\pm2.8}$ | $82.3_{\pm1.9}$ | $57.3_{\pm2.4}$ | $52.8_{\pm1.7}$ | $54.5_{\pm1.8}$ | 66.31 |
| | **Trans-Encoder** | $67.9_{\pm3.8}$ | $70.9_{\pm4.1}$ | $75.2_{\pm8.8}$ | $77.7_{\pm2.5}$ | $61.4_{\pm6.0}$ | $56.7_{\pm1.7}$ | $82.8_{\pm2.0}$ | $57.3_{\pm4.3}$ | $52.7_{\pm2.0}$ | $59.3_{\pm3.9}$ | 66.21 |
| | **$k$NN Prompting** | $87.7_{\pm3.5}$ | $77.0_{\pm3.5}$ | $75.2_{\pm8.8}$ | $86.2_{\pm1.8}$ | $58.9_{\pm2.2}$ | $88.2_{\pm3.5}$ | $94.1_{\pm2.3}$ | $83.9_{\pm2.4}$ | $53.6_{\pm3.0}$ | $64.8_{\pm4.2}$ | **76.97** |
| | **BM25** | $69.7_{\pm2.9}$ | $67.7_{\pm2.5}$ | $79.4_{\pm2.0}$ | $71.7_{\pm1.9}$ | $68.6_{\pm4.1}$ | $54.7_{\pm1.1}$ | $83.4_{\pm2.0}$ | $58.7_{\pm1.5}$ | $52.0_{\pm1.6}$ | $65.9_{\pm5.6}$ | 67.18 |
| | **SBERT** | $71.8_{\pm3.4}$ | $71.6_{\pm3.8}$ | $80.2_{\pm1.5}$ | $84.6_{\pm2.2}$ | $66.8_{\pm6.8}$ | $59.8_{\pm1.8}$ | $84.6_{\pm0.8}$ | $57.6_{\pm1.6}$ | $52.8_{\pm3.6}$ | $63.7_{\pm4.7}$ | 69.37 |
| $m = 64$ | **SimCSE** | $68.4_{\pm4.7}$ | $69.7_{\pm2.6}$ | $81.6_{\pm0.6}$ | $83.1_{\pm2.2}$ | $71.4_{\pm4.6}$ | $57.9_{\pm2.6}$ | $84.8_{\pm2.1}$ | $57.3_{\pm2.0}$ | $52.3_{\pm3.6}$ | $58.1_{\pm1.8}$ | 68.46 |
| | **Trans-Encoder** | $69.3_{\pm4.3}$ | $73.0_{\pm1.2}$ | $82.0_{\pm2.2}$ | $79.3_{\pm2.0}$ | $69.3_{\pm3.4}$ | $57.9_{\pm2.7}$ | $85.6_{\pm1.2}$ | $57.9_{\pm1.4}$ | $52.1_{\pm4.5}$ | $60.1_{\pm2.6}$ | 68.65 |
| | **$k$NN Prompting** | $87.2_{\pm2.8}$ | $82.5_{\pm1.2}$ | $83.0_{\pm4.0}$ | $86.2_{\pm1.1}$ | $62.9_{\pm6.1}$ | $89.1_{\pm1.6}$ | $95.5_{\pm1.4}$ | $84.3_{\pm1.9}$ | $49.1_{\pm3.2}$ | $68.9_{\pm2.9}$ | **78.86** |
| | **BM25** | $69.1_{\pm0.5}$ | $66.8_{\pm2.7}$ | $75.2_{\pm6.2}$ | $77.5_{\pm1.4}$ | $71.4_{\pm0.0}$ | $56.4_{\pm1.2}$ | $85.5_{\pm1.7}$ | $59.8_{\pm2.0}$ | $54.5_{\pm1.5}$ | $72.3_{\pm4.9}$ | 68.87 |
| | **SBERT** | $71.7_{\pm1.9}$ | $71.9_{\pm1.7}$ | $79.6_{\pm3.6}$ | $85.3_{\pm2.0}$ | $69.6_{\pm0.0}$ | $58.8_{\pm0.8}$ | $87.2_{\pm0.9}$ | $60.1_{\pm2.3}$ | $53.3_{\pm2.3}$ | $64.9_{\pm2.1}$ | 70.24 |
| $m = 128$ | **SimCSE** | $70.9_{\pm2.2}$ | $71.6_{\pm3.2}$ | $81.6_{\pm2.6}$ | $84.5_{\pm1.7}$ | $73.2_{\pm0.0}$ | $58.5_{\pm2.2}$ | $87.0_{\pm2.1}$ | $58.7_{\pm1.4}$ | $53.4_{\pm3.2}$ | $59.6_{\pm3.3}$ | 69.89 |
| | **Trans-Encoder** | $69.0_{\pm1.3}$ | $75.5_{\pm2.2}$ | $82.6_{\pm1.9}$ | $82.9_{\pm0.6}$ | $73.2_{\pm0.0}$ | $56.1_{\pm0.8}$ | $87.0_{\pm1.4}$ | $57.1_{\pm1.3}$ | $52.9_{\pm3.1}$ | $63.9_{\pm2.3}$ | 70.02 |
| | **$k$NN Prompting** | $86.3_{\pm2.9}$ | $83.8_{\pm2.1}$ | $82.3_{\pm1.5}$ | $87.2_{\pm0.4}$ | $64.6_{\pm7.9}$ | $88.9_{\pm2.4}$ | $96.5_{\pm0.7}$ | $86.4_{\pm0.8}$ | $51.1_{\pm1.7}$ | $74.0_{\pm2.9}$ | **80.12** |
| | **BM25** | $72.0_{\pm3.9}$ | $72.3_{\pm1.4}$ | $78.8_{\pm3.2}$ | $77.3_{\pm3.0}$ | $71.4_{\pm0.0}$ | $57.1_{\pm0.8}$ | $88.2_{\pm1.9}$ | $58.4_{\pm1.6}$ | $53.8_{\pm2.8}$ | $76.6_{\pm3.1}$ | 70.60 |
| | **SBERT** | $69.9_{\pm1.8}$ | $72.3_{\pm0.8}$ | $82.2_{\pm2.5}$ | $86.3_{\pm1.2}$ | $69.6_{\pm0.0}$ | $58.8_{\pm1.5}$ | $88.5_{\pm0.9}$ | $59.8_{\pm2.6}$ | $52.3_{\pm2.4}$ | $69.1_{\pm0.8}$ | 70.89 |
| $m = 256$ | **SimCSE** | $71.4_{\pm3.7}$ | $73.7_{\pm1.6}$ | $82.9_{\pm0.8}$ | $85.5_{\pm1.4}$ | $73.2_{\pm0.0}$ | $59.7_{\pm1.2}$ | $89.1_{\pm1.7}$ | $58.8_{\pm2.2}$ | $51.1_{\pm2.6}$ | $64.3_{\pm1.5}$ | 70.87 |
| | **Trans-Encoder** | $70.0_{\pm1.0}$ | $76.6_{\pm1.4}$ | $82.1_{\pm2.0}$ | $84.1_{\pm1.2}$ | $73.2_{\pm0.0}$ | $58.0_{\pm1.0}$ | $89.9_{\pm1.3}$ | $58.1_{\pm1.2}$ | $52.0_{\pm2.4}$ | $70.9_{\pm2.2}$ | 71.49 |
| | **$k$NN Prompting** | $89.1_{\pm1.6}$ | $85.0_{\pm2.2}$ | $83.4_{\pm1.5}$ | $87.5_{\pm1.7}$ | $64.6_{\pm7.9}$ | $89.3_{\pm0.9}$ | $97.3_{\pm0.3}$ | $85.6_{\pm2.0}$ | $52.7_{\pm3.3}$ | $78.1_{\pm3.3}$ | **81.27** |
| | **BM25** | $74.6_{\pm2.0}$ | $71.8_{\pm1.3}$ | $79.0_{\pm2.3}$ | $81.3_{\pm1.7}$ | $71.4_{\pm0.0}$ | $58.4_{\pm1.1}$ | $88.7_{\pm1.9}$ | $59.7_{\pm1.1}$ | $53.7_{\pm2.3}$ | $82.9_{\pm1.1}$ | 72.14 |
| | **SBERT** | $71.6_{\pm2.8}$ | $74.3_{\pm1.2}$ | $83.1_{\pm2.2}$ | $89.1_{\pm1.9}$ | $69.6_{\pm0.0}$ | $59.2_{\pm3.0}$ | $88.6_{\pm1.5}$ | $58.8_{\pm2.1}$ | $51.8_{\pm3.0}$ | $73.8_{\pm1.9}$ | 72.00 |
| $m = 512$ | **SimCSE** | $73.6_{\pm2.0}$ | $75.5_{\pm3.1}$ | $83.9_{\pm1.6}$ | $86.6_{\pm1.6}$ | $73.2_{\pm0.0}$ | $59.1_{\pm1.0}$ | $90.4_{\pm1.3}$ | $59.2_{\pm2.4}$ | $50.2_{\pm2.3}$ | $69.1_{\pm2.1}$ | 72.07 |
| | **Trans-Encoder** | $73.1_{\pm3.6}$ | $78.9_{\pm1.7}$ | $84.2_{\pm1.1}$ | $87.1_{\pm1.7}$ | $73.2_{\pm0.0}$ | $57.7_{\pm1.1}$ | $89.6_{\pm1.2}$ | $58.4_{\pm2.5}$ | $52.1_{\pm3.7}$ | $74.1_{\pm2.3}$ | 72.84 |
| | **$k$NN Prompting** | $89.0_{\pm1.5}$ | $87.6_{\pm1.6}$ | $83.7_{\pm1.2}$ | $88.4_{\pm1.5}$ | $64.6_{\pm7.9}$ | $90.1_{\pm1.0}$ | $97.6_{\pm0.5}$ | $85.1_{\pm1.6}$ | $52.9_{\pm1.8}$ | $80.5_{\pm1.8}$ | **81.94** |
| | **BM25** | $76.9_{\pm1.8}$ | $74.9_{\pm2.7}$ | $79.3_{\pm2.2}$ | $83.6_{\pm1.3}$ | $71.4_{\pm0.0}$ | $56.9_{\pm1.0}$ | $90.2_{\pm1.6}$ | $60.3_{\pm2.7}$ | $52.7_{\pm1.0}$ | $85.0_{\pm1.2}$ | 73.12 |
| | **SBERT** | $73.9_{\pm1.6}$ | $76.6_{\pm2.3}$ | $83.7_{\pm1.2}$ | $90.0_{\pm1.6}$ | $69.6_{\pm0.0}$ | $58.1_{\pm1.1}$ | $91.1_{\pm1.2}$ | $59.6_{\pm0.9}$ | $52.8_{\pm1.8}$ | $75.7_{\pm2.0}$ | 73.12 |
| $m = 1024$ | **SimCSE** | $73.5_{\pm3.0}$ | $74.5_{\pm1.8}$ | $84.6_{\pm1.1}$ | $88.7_{\pm1.1}$ | $73.2_{\pm0.0}$ | $57.5_{\pm1.0}$ | $91.6_{\pm0.8}$ | $59.6_{\pm1.3}$ | $53.1_{\pm1.0}$ | $69.4_{\pm1.3}$ | 72.59 |
| | **Trans-Encoder** | $73.4_{\pm2.2}$ | $78.4_{\pm1.6}$ | $82.9_{\pm0.8}$ | $87.8_{\pm2.3}$ | $73.2_{\pm0.0}$ | $55.9_{\pm0.8}$ | $91.7_{\pm0.8}$ | $59.5_{\pm3.2}$ | $54.8_{\pm2.0}$ | $76.6_{\pm0.7}$ | 73.42 |
| | **$k$NN Prompting** | $88.2_{\pm1.0}$ | $88.4_{\pm1.8}$ | $84.1_{\pm1.2}$ | $89.1_{\pm1.2}$ | $64.6_{\pm7.9}$ | $86.7_{\pm1.3}$ | $97.8_{\pm0.6}$ | $84.4_{\pm1.5}$ | $53.3_{\pm4.2}$ | $83.0_{\pm1.4}$ | **81.96** |

Table 9: Full results for comparison to PromptCompose, corresponding to Figure 6. Some ICL results are reused (MPQA, 32 shot), see footnote in caption of Table 8 for explanation.

## C.4 Benefits of Whole LM Distribution Under Fully Supervised Scenario

We provide comparison between whole and partial LM distribution also on fully supervised scenario in Table 10. The conclusion remains invariant with few shot scenario as demonstrated in Table 2.

| Methods | SST2 | SUBJ | MPQA | AGNews | CB | CR | DBPedia | MR | RTE | TREC | AVG |
|---|---|---|---|---|---|---|---|---|---|---|---|
| **ICL** | $81.3_{\pm5.4}$ | $64.1_{\pm11.3}$ | $75.2_{\pm8.8}$ | $72.7_{\pm18.5}$ | $60.7_{\pm2.8}$ | $66.2_{\pm16.7}$ | $83.5_{\pm3.8}$ | $72.2_{\pm13.9}$ | $53.0_{\pm1.7}$ | $54.2_{\pm4.9}$ | 68.31 |
| **$k$NN Prompting** | $88.2_{\pm1.0}$ | $88.4_{\pm1.8}$ | $84.1_{\pm1.2}$ | $89.1_{\pm1.2}$ | $64.6_{\pm7.9}$ | $86.7_{\pm1.3}$ | $97.8_{\pm0.6}$ | $84.4_{\pm1.5}$ | $53.3_{\pm4.2}$ | $83.0_{\pm1.4}$ | **81.96** |
| **$k$NN Prompting (Partial)** | $87.7_{\pm1.4}$ | $69.3_{\pm8.9}$ | $83.4_{\pm1.1}$ | $84.5_{\pm1.2}$ | $63.2_{\pm6.0}$ | $84.1_{\pm2.7}$ | $96.8_{\pm0.6}$ | $84.7_{\pm1.8}$ | $49.0_{\pm2.9}$ | $65.7_{\pm3.5}$ | 76.84 |

Table 10: Comparison between whole and partial LM distribution on fully supervised scenario.

---

[7] In Table 8, there are few cases where $k$NN Prompting gets identical results with ICL baseline. This happens when $M_T$=32, which leaves $|\mathcal{A}| = 0$ as we have invariably set $|\mathcal{D}| = M_T$. To avoid exploiting exceptional split strategies for such specific case, we simply re-use the Underperformed ICL baseline results. In general, this only occurs on MPQA (when $m = 32$) and SST2 (when $m = 32$ and LLM > 2.7B, 2048 tokens context), and should have few impact on the overall results and conclusion. Similar situation happens in Table 9

[8] `https://huggingface.co/sentence-transformers/all-MiniLM-L6-v2`

[9] `https://huggingface.co/cambridgeltl/trans-encoder-bi-simcse-bert-base`

[10] `https://huggingface.co/princeton-nlp/unsup-simcse-bert-base-uncased`

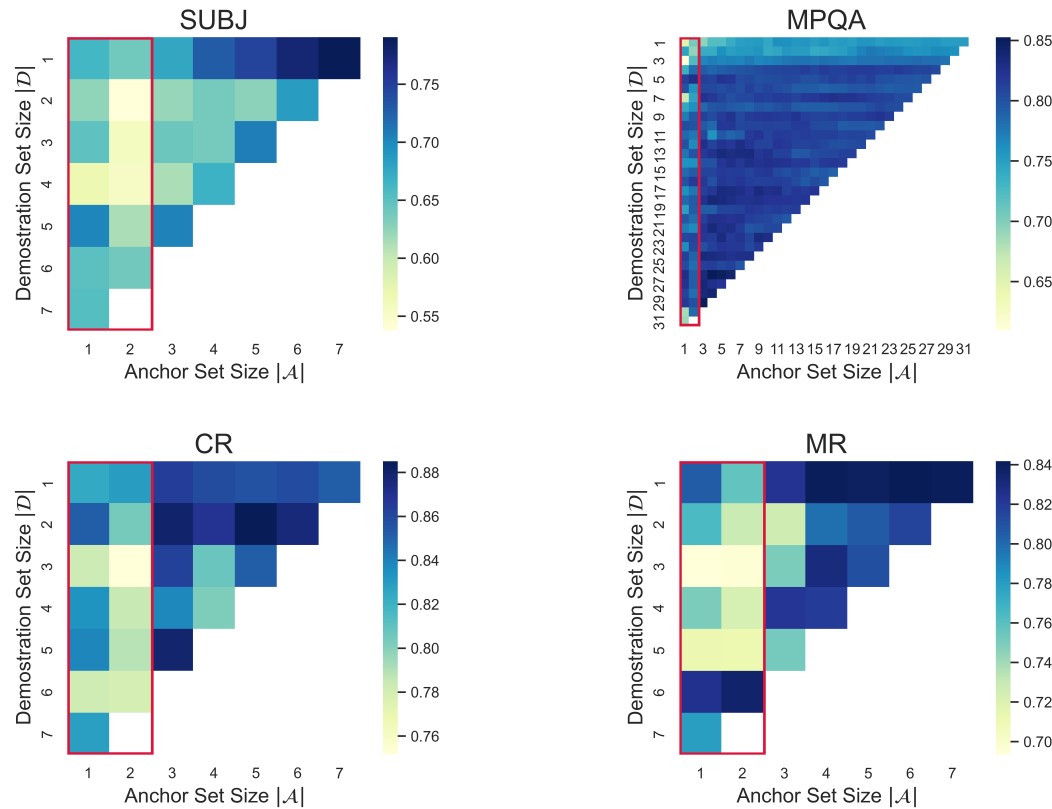

Figure 13: Split of Demonstration and Anchor Set. $|\mathcal{A}| + |\mathcal{D}| \leq M_T$.

## C.5 SPLIT STRATEGY OF DEMONSTRATION AND ANCHOR SET

We provide results on more datasets regarding the investigation of split strategy in Figure 13, corresponding to Section 4.3.2.

## D COMPARISON TO FINETUNING UNDER FULLY SUPERVISED SCENARIO

We compare $k$NN Prompting to standard PLM finetuning in a more extensive data scale. For finetuning baselines in both Table 3 and Figure 14, we set hyper-parameters following previous works (Schick & Schütze, 2021). We set learning rate to 1e-5, batch size to 16, and training steps to 125, 250 or 500, respectively for $m \in \{32, 64\}, \{128, 256\}, \{512, 1024\}$. For CB, AGNews and RTE, batch size is adjusted to 8, for DBPedia, batch size is adjusted to 4 to avoid OOM. We observe that with the same model scale, $k$NN Prompting is superior than finetuning under the low resource setting, but inferior under fully supervised setting. This indicates its data utility factor $\alpha$ is still smaller than finetuning. However, the main advantage of $k$NN Prompting comes with LLM, which significantly outperforms the finetuning baseline without any gradient-based optimization. We have also discussed this in Section 6.

## E MORE CASE STUDY

We provide more case study, respectively on SST2, TREC, AGNews and MR (Figure 15, 17, 16 and 18).

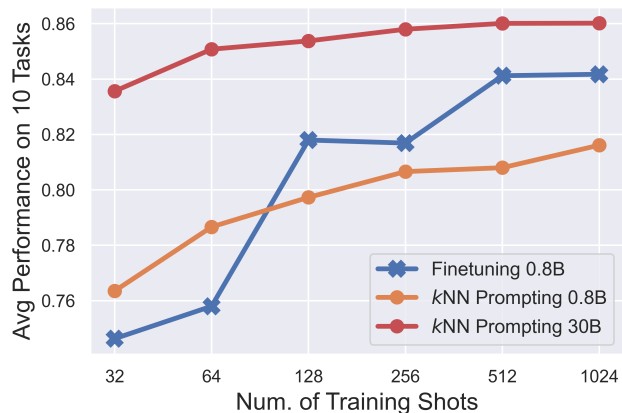

Figure 14: Comparison to finetuning baseline.

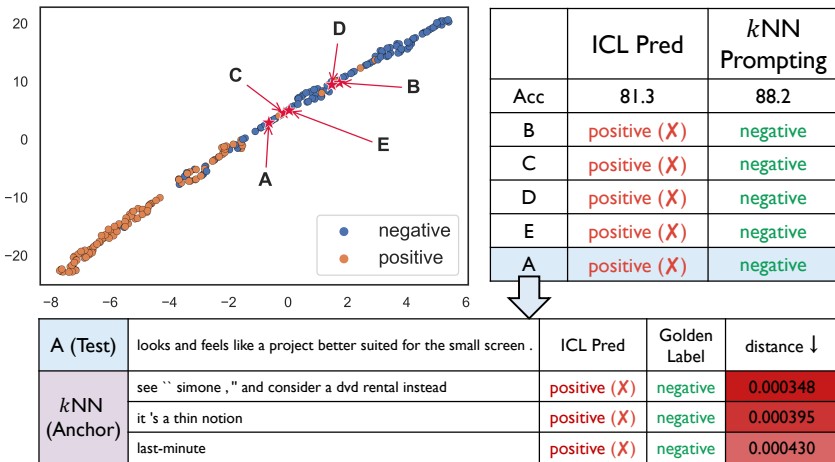

Figure 15: Case study on SST2, all 5 test instances are randomly selected given that $k$NN Prompting outperforms ICL.

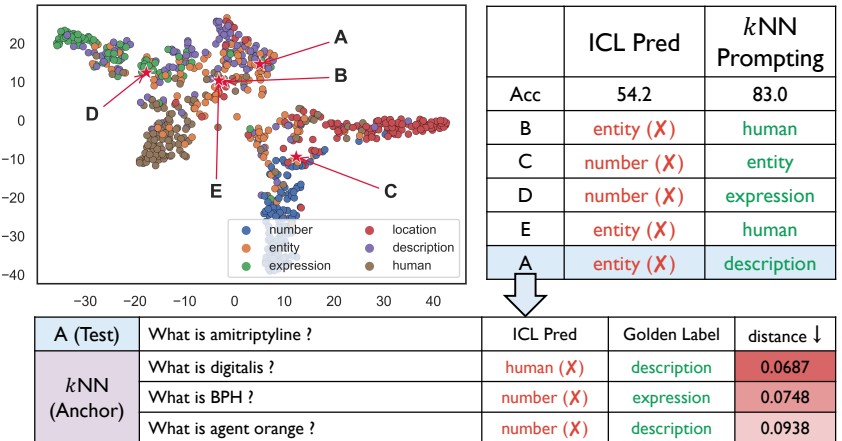

Figure 16: Case study on TREC, all 5 test instances are randomly selected given that $k$NN Prompting outperforms ICL.

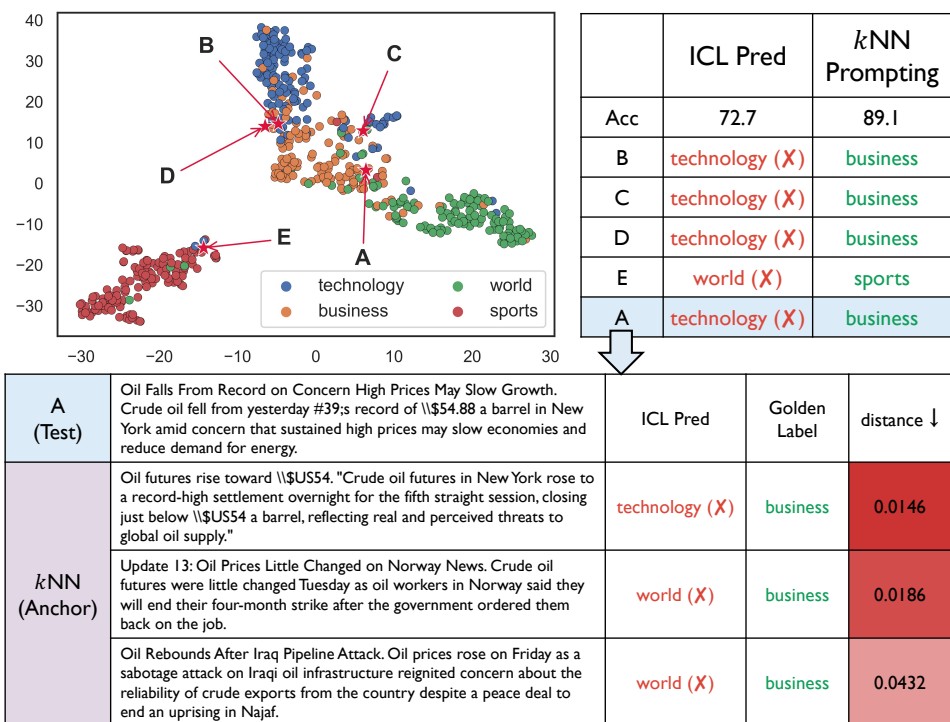

Figure 17: Case study on AGNews, all 5 test instances are randomly selected given that $k$NN Prompting outperforms ICL.

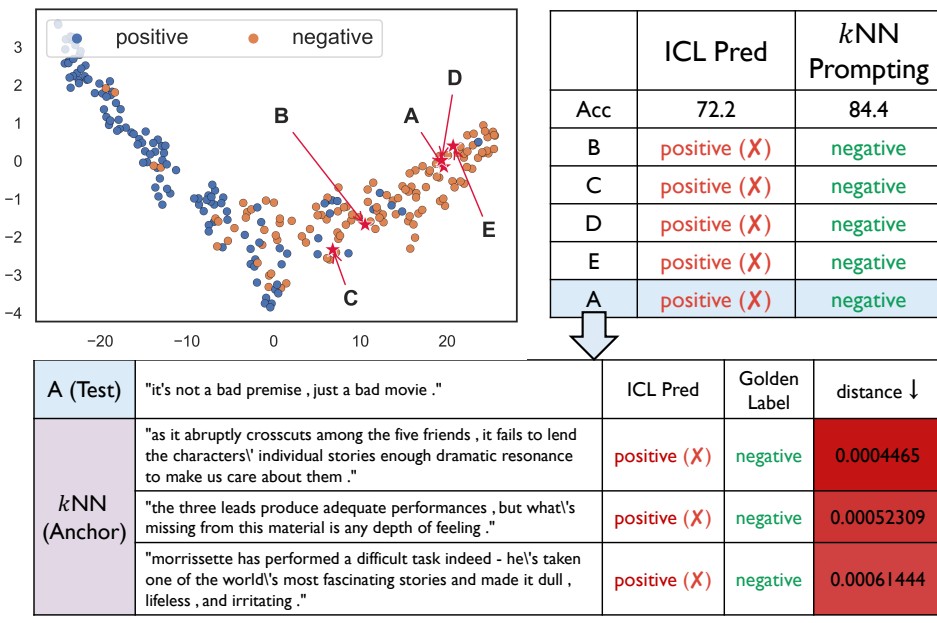

Figure 18: Case study on MR, all 5 test instances are randomly selected given that $k$NN Prompting outperforms ICL.

## F PROMPT TEMPLATE

See Table 11 for the used templates (Adopted from Lu et al. (2022)). They are intuitively designed and the proposed method should be robust with choices of templates.

| Task | Template | Label Space |
|---|---|---|
| SST2 | Review: contains no wit , only labored gags
Sentiment: negative
Review: the film is powerful , accessible and funny .
Sentiment: | negative, positive |
| SUBJ | Input: the script isn't very good ; not even someone as gifted as hoffman ( the actor ) can make it work .
Type: subjective
Input: he must do this in secret so that the parents and school personnel know nothing of his plan .
Type: | subjective, objective |
| MPQA | Review: would not find it at all strange
Sentiment: negative
Review: as small ( yet acceptable ) as possible
Sentiment: | negative, positive |
| AGNews | Input: Carlyle Looks Toward Commercial Aerospace (Reuters). "Reuters - Private investment firm Carlyle Group, which has a reputation for making well-timed and occasionally controversial plays in the defense industry, has quietly placed its bets on another part of the market.
Type: technology
Input: Superstar Kewell remains centre of attention. Socceroo forward Harry Kewell loosens up by tossing around a ball at Bondi beach yesterday. Photo: Craig Golding. There were half a dozen Socceroos standing on a raised platform in Sydney #39;s
Type: | world, sports, business, technology |
| CB | Premise: It was a complex language. Not written down but handed down. One might say it was peeled down.
Hypothesis: the language was peeled down
Prediction: False
Premise: A: so I don't know if I wasn't drug tested based on that or because the man who hired me didn't request the drug test, because I know that my company does drug testing on occasion. B: Right. Well, for instance, does the company you worked for before have the right or do they have the ability to say, hey, we've already drug tested her and she came up negative. A: Well, no, I don't think they can force another company to not drug test me just by saying that I didn't, I mean,
Hypothesis: they can force another company to not drug test her
Prediction: | False, True, Neither |
| CR | Review: it 's not as stylized as a sony or samsung .
Sentiment: negative
Review: i went out and got the canon today .
Sentiment: | negative, positive |
| DBPedia | Input: Geoffrey D. Falksen (born July 31 1982) is an American steampunk writer.
Type: artist
Input: Monster Night is a 2006 film directed by Leslie Allen and Lorenzo Doumani.
Type: | company, school, artist, athlete, politics, transportation, building, nature, village, animal, plant, album, film, book |
| MR | Review: "you might say tykwer has done all that heaven allows , if you wanted to make as anti-kieslowski a pun as possible . suffice to say its total promise is left slightly unfulfilled ."
Sentiment: negative
Review: an alternately raucous and sappy ethnic sitcom . . . you'd be wise to send your regrets .
Sentiment: | negative, positive |
| RTE | Premise: A man is due in court later charged with the murder 26 years ago of a teenager whose case was the first to be featured on BBC One's Crimewatch. Colette Aram, 16, was walking to her boyfriend's house in Keyworth, Nottinghamshire, on 30 October 1983 when she disappeared. Her body was later found in a field close to her home. Paul Stewart Hutchinson, 50, has been charged with murder and is due before Nottingham magistrates later."
Hypothesis: Paul Stewart Hutchinson is accused of having stabbed a girl.
Prediction: false
Premise: For women earning 22,000 a year, the total pay accumulated after six months maternity leave would be just 5,300 in the UK and 5,850 in Ireland. Entitlements in Germany would also be relatively low, at 5,900, along with those in France, Spain and the Netherlands, all at 6,750. At the other end of the scale, pay received after six months leave in Italy would be 9,150 while in Denmark and Norway it would be as much as 11,000.
Hypothesis: Maternity leave varies in Europe.
Prediction: | false, true |
| TREC | Question: How did serfdom develop in and then leave Russia ?
Type: description
Question: What is Shakespeare 's nickname ?
Type: | description, entity, expression, human, location, number |

Table 11: Templates for ICL. These are minimum cases with only one demonstration example for illustration.

