# OpenReview forum: "$k$NN Prompting: Beyond-Context Learning with Calibration-Free Nearest Neighbor Inference"
_ICLR.cc/2023/Conference — ICLR 2023 poster_

### Official Review · Reviewer_SRJJ · 2022-10-25

**Confidence:** 4
**Correctness:** 4
**Technical Novelty And Significance:** 4
**Empirical Novelty And Significance:** 3
**Recommendation:** 6

**Clarity, Quality, Novelty And Reproducibility:**

Novelty: the major novelty of this paper is the proposal of kNN style inference over the representations of the training data. It opens an interesting direction for the efficient deployment of LLM.

Quality: this paper is well-organized and well-presented, with a clear introduction to the kNN prompting.


**Strength And Weaknesses:**

The kNN prompting idea is novel. It aligns with the retrieval philosophy that we can make full use of the trained large language models for a better inference. The strength can be summarized as follows:

1. The kNN prompting is a gradient-free paradigm for deploying LLM.

2. The kNN prompting integrates the idea of the nearest neighbor in the inference of LLM, improving the LLM performance without further tuning.

3. The empirical results are promising, especially for long sequences.

**Summary Of The Paper:**

This paper proposes kNN prompting, a novel retrieval-based model for more accurate inference accuracy. kNN prompting queries LLM with the training data and get the representation as anchors. In the prediction phase, kNN prompting only performs nearest neighbor for a better accruacy.

**Summary Of The Review:**

This paper proposes a novel idea for deploying large-scale LLM in practice. This paper is also well-written with clear illustration of kNN prompting.  The empirical evaluation also suggest the strength of kNN prompting in long sequences.

---

> ### Author Response · Authors · 2022-11-19
> **Answers to Review SRJJ**
>
> Thanks for the positive evaluation including novel idea, promising results and well-presented!\
> We have now further refined the Abstract and Introduction to strengthen and clarify our contribution. We have also added new experiments and analyses to cover a more comprehensive scope. We kindly refer the reviewer to these revisions, and please let us know if there are any follow-up questions, we will try our best to answer them as soon as possible.

---

### Official Review · Reviewer_vijR · 2022-10-25

**Confidence:** 4
**Correctness:** 3
**Technical Novelty And Significance:** 3
**Empirical Novelty And Significance:** 3
**Recommendation:** 6

**Clarity, Quality, Novelty And Reproducibility:**

In terms of presentations, the paper is overall well-written except:

1. In Table 2, the results for MPQA with m=32 are the exactly same for ICL and kNN prompting; such typos also appear in other parts of the table.

2. The notations introduced in Section 3, which is used throughout the paper, are heavier than need and could be further simplified: for example, the authors may want to remove \theta from the formulation and the subscripts are a little bit messy.

The authors provide enough details of experiments for reproducing the results.

**Strength And Weaknesses:**

Strengths:
kNN prompting is a simple yet very effective approach for scaling LLMs inference to large training sets; comprehensive ablation study demonstrates the robustness of the approach across different choices of hyper parameters (k) and random seeds.

Weaknesses:
The discussion (Sec. 4.3.5) on why kNN prompting outperforms vanilla ICL in a few-shot setting is very interesting: it seems to suggest that kNN prompting is implicitly solving the issue of “surface form competition” where some answers have higher probabilities in the prior distribution of LLM. I wonder how kNN prompting compares against to other calibration techniques [3] that solves the same problem.

In terms of novelty, the method itself is more of a variation of the methods proposed in prior works that retrieve extra data via kNN to help model inference [1, 2].

[1] Urvashi Khandelwal, Omer Levy, Dan Jurafsky, Luke Zettlemoyer, and Mike Lewis. Generalization through memorization: Nearest neighbor language models. In International Conference on Learn- ing Representations, 2020. URL https://openreview.net/forum?id=HklBjCEKvH.
[2] Weijia Shi, Julian Michael, Suchin Gururangan, and Luke Zettlemoyer. Nearest neighbor zero-shot inference. arXiv preprint arXiv:2205.13792, 2022.
[3] Min, Sewon, Mike Lewis, Hannaneh Hajishirzi, and Luke Zettlemoyer. "Noisy channel language model prompting for few-shot text classification." arXiv preprint arXiv:2108.04106 (2021).


**Summary Of The Paper:**

In-context Learning (ICL) is one of the most prevailing paradigms for inference with large language models (LLM) without parameter update. However, as the sizes of context windows are limited for LLMs, only a small number of training examples can be used for prompting, preventing ICL from scaling to larger training set. To overcome this bottleneck, this paper proposed a new method for LLM inference with no parameter update called kNN prompting: unlike ICL, where labels are directly predicted based on next-token distribution, kNN prompting make predictions by leveraging/retrieving training examples (when available) that share similar next-token distributions with the test example. Empirical results show that kNN prompting not only scales effectively as the number of training examples increase but also outperforms vanilla ICL on some datasets in the few-shot setting. Moreover, when a relatively large training set is available, kNN prompting even outperforms the (relatively smaller) fine-tuned models as the number of parameters of LLMs exceed a certain threshold (6B). To summarize, this paper proposed a simple yet effective way of scaling LLMs inference to large training sets with no parameter update.


**Summary Of The Review:**

Overall this is a good paper that proposes a simple and elegant approach with strong empirical results. The only weakness is the limited novelty of the methodology itself; to provide more unique insights, the authors may want to provide a more detailed analysis about the question discussed in 4.3.5.

---

> ### Author Response · Authors · 2022-11-19
> **Answers to Review vijR (1/2)**
>
> We thank for the reviewer's positive evaluation such as simple, very effective, elegant, strong results, etc. There are also very valuable questions and advices we try to answer as follows.
>
> > The discussion (Sec. 4.3.5) on why kNN prompting outperforms vanilla ICL in a few-shot setting is very interesting: it seems to suggest that kNN prompting is implicitly solving the issue of “surface form competition” where some answers have higher probabilities in the prior distribution of LLM. I wonder how kNN prompting compares against to other calibration techniques [1] that solves the same problem.
>
> This is a very insightful opinion! As LLM is trained on general-domain corpora, it's capability to complete a fabricated prompt thus is essentially not well aligned with downstream task-specific label space, and inevitably suffers from various biases, including the “*surface form competition*”.\
> However, $k$NN Prompting takes an entirely different road, instead of trying to calibrate the input-label alignment, it leverages LLM representation to align test and training instances, and refer to the golden label of anchors to make predictions, thus is **calibration-free**. From such a perspective, it indeed implicitly resolves the “*surface form competition*” problem.\
> We have now strengthened such a motivation in Introduciton and expanded the discussion in Section 4.3.3.
>
> To demonstrate the superiority of the proposed calibration-free optimization, we have added comparisons to state-of-the-art calibration-based methods [1][2] as the reviewer instructed. The comparison is performed under strictly comparable few shot scenario (though $k$NN Prompting can further scale up), the results can well support the above claim and conclusion. We excerpt some of the results below, and kindly refer the reviewer to comprehensive results in Section 4.2.1 Table 2 and the visualized scaling curve in Section 4.2.1 Figure 4 and Appendix C.1 Figure 12.
>
> |Method|SST2|SUBJ|MPQA|MR|CR|TREC|AVG|
> |--|--|--|--|--|--|--|--|
> |$m=2$ (Training Data)|
> |In-Context Learning|59.8|51.4|60.2|57.3|64.7|50.0|57.24|
> |Contextual Calibration[2]|76.2|69.8|63.4|60.1|75.6|45.5|65.12|
> |Noisy Channel[1]|82.6|64.6|60.2|83.3|79.4|35.2|**67.54**|
> |$k$NN Prompting|77.5|73.4|56.6|69.7|81.1|41.3|66.57 (-0.97)|
> |$m=4$ (Training Data)|
> |In-Context Learning|67.2|56.5|70.8|60.8|62.8|50.9|61.50|
> |Contextual Calibration|70.8|60.0|70.5|59.6|70.0|43.6|62.43|
> |Noisy Channel|80.9|60.5|66.4|83.9|79.0|40.9|68.58|
> |$k$NN Prompting|87.1|73.5|66.4|71.2|82.9|51.6|**72.14 (+3.29)**|
> |$m=8$ (Training Data)|
> |In-Context Learning|57.8|66.2|77.2|66.0|61.5|50.9|63.27|
> |Contextual Calibration|68.5|64.5|72.7|64.9|68.2|44.0|63.80|
> |Noisy Channel|82.0|62.5|70.1|85.0|79.2|41.6|70.06|
> |$k$NN Prompting|88.9|77.7|72.5|75.4|84.6|63.7|**77.13 (+7.07)**|
>
> > In terms of novelty, the method itself is more of a variation of the methods proposed in prior works that retrieve extra data via kNN to help model inference.
>
> These are definitely very related works as cited and discussed in the paper. To a more broad scope, KNN is in itself a very classic idea that impacted a wide range of areas. This work novelly adapts this classic technique into the LLM inference paradigm, achieves state-of-the-art few-shot performance as well as beyond-context data scaling capability. The novelty lies in the important scenario where the technique is introduced, the effectiveness it enables, as well as potential changes and insights it inspires.
>
> > In Table 2, the results for MPQA with $m=32$ are the exactly same for ICL and kNN prompting; such typos also appear in other parts of the table.
>
> Thanks for the careful examination! This is expected and caused by our compromised experimental configuration for anchor-demonstration set split: under low-resource and fully supervised scenario, we **invariably** split demonstration set size $|\mathcal{D}|=M_T$ (maximum allowed shots). On the specific case of MPQA with $m=32$, we thus have $m=M_T=|\mathcal{D}|$, leaving anchor set size $|\mathcal{A}|=0$, so we directly re-use the underperformed ICL baseline results to avoid exploiting exceptional split strategies, which otherwise will actually improve the reported performance. We have clarified this detail in caption of Appendix C Table 7 & 8.\
> In general, this only occurs on MPQA (when $m=32$) and SST2 (when $m=32$ *and* LLM > 2.7B, 2048 tokens context), and should have few impacts on the overall results and conclusion.\
> In the revised version, $m=32$ setting is also added the in few shot scenario (Section 4.2.1 Table 2), where we **invariably** set $|\mathcal{D}|=1$, which could provide discriminative results for potential future reference.

---

> > ### Author Response · Authors · 2022-11-19
> > **Answers to Review vijR (2/2)**
> >
> > > The notations introduced in Section 3, which is used throughout the paper, are heavier than need and could be further simplified...
> >
> > Thanks for the detailed advice! We have tried our best to simplify the notations in the revised version, e.g., the subscript $k_{a_i}$ is simplified to $k_i$, distribution $p_{infer}$ is written as $p_{test}$ to be consistent with $P_{test}$ and $x_{test}$, $\mathcal{V}$ and $\mathcal{M}$ are discarded, etc. Hope this will make the presentation easier to read.
> >
> > [1] Noisy Channel Language Model Prompting for Few-Shot Text Classification. (Min et al., ACL 2022)\
> > [2] Calibrate before use: Improving few-shot performance of language models. (Zhao et al., ICML 2021)

---

### Official Review · Reviewer_wqbJ · 2022-10-29

**Confidence:** 4
**Correctness:** 4
**Technical Novelty And Significance:** 3
**Empirical Novelty And Significance:** 3
**Recommendation:** 8

**Clarity, Quality, Novelty And Reproducibility:**

Clarity & Quality: the paper is generally well-written and easy to follow. One thing confusing me is Table 1 -- I did not see authors mentioning Table 1 and the numbers in Table 1 seem contradictory to other tables.

Novelty: The method/idea is built based on existing retrieval-based models. The technical novelty is limited -- however, the paper provides insightful and interesting results to the field.

Reproducibility: The results should not be hard to reproduce, especially the authors claim they will release the code.


**Strength And Weaknesses:**

Strengths:
- In general, the paper proposes a simple and effective solution to enable the prompting methods to leverage more training examples, instead of merely relying on the demonstrations in the context. The approach is very interesting and the results are strong.
- The motivation of the paper is clear and makes sense to me; and the paper is positioned well. I appreciate the authors' efforts on addressing the sequence-length issue when using ICL. I believe that this is a crucial problem of using LLMs on downstream tasks in a full supervision setting and the community should facilitate addressing this problem.
- The paper is presented clearly.
- I also like the discussion section where the authors raise potential concerns for the existing retrieval-based models. I believe addressing these problems can further reveal the capacity of retrieval-based models.
- The paper has conducted ablation studies for understanding the model designs, e.g., KL divergence vs L2 distance; the sensitivity of the number of demonstrations/anchors in kNN Prompting. These studies are informative.


Weaknesses:
- What confuses me is that during inference of kNN Prompting, the method will retrieve anchor examples that have similar predicted probability distributions (measured by KL divergence). Intuitively, this means that the predictions are likely to be the same (or at least very similar) to the current testing example. That means it is not likely that the prediction will be different. I expect detailed discussion about how retrieving anchor examples can actually help the predictions.
- The predictions are made based on majority voting of retrieved examples. This method may suffer when the labels are imbalanced. For example, when one label has much fewer examples compared to other labels, this label will not be favored by the majority voting system.

**Summary Of The Paper:**

This paper proposes kNN Prompting, which addresses the problem of in-context learning (ICL) that the context length is limited, so that the performance cannot scale with the number of available training examples. Similar to ICL, kNN Prompting also formulates the downstream tasks as prompt completion tasks. Differently, at inference time, kNN Prompting not only relies on the demonstration examples in the context of the input, but also will retrieve anchor examples from a datastore built by taking all the training examples.

The paper conducts experiments on ten downstream tasks, with different model sizes and different scales of training set. They find kNN Prompting outperform ICL under few shot settings and can scale up with as many training examples as are available. Ablation studies and analyses are conducted to study the model design choices.

**Summary Of The Review:**

In summary, I think this paper is well-motivated and is written clearly. It presents very interesting and insightful results. I hold a positive attitude towards accepting this paper.

---

> ### Author Response · Authors · 2022-11-19
> **Answers to Review wqbJ**
>
> We thank the reviewer for such a positive evaluation including interesting and insightful, strong results, well-positioned, etc, and the agreement on the cruciality of the addressed problem. We answer the questions as follows.
>
> > What confuses me is that during inference of kNN Prompting, the method will retrieve anchor examples that have similar predicted probability distributions (measured by KL divergence). Intuitively, this means that the predictions are likely to be the same (or at least very similar) to the current testing example. That means it is not likely that the prediction will be different. I expect detailed discussion about how retrieving anchor examples can actually help the predictions.
>
> This is a very thought-provoking question, and we try to explain as follows:\
> Indeed the retrieved anchors tend to have similar LM probability with the test instance, and accordingly similar label predictions. As illustrated in Figure 9, the test example and its top 2 nearest anchors are all predicted the same.\
> However, although LM probability itself is **inferior** for making predictions, they are **superior** for matching examples because it entails delicate representations generated by LLM.
> And $k$NN Prompting exactly exploits the LM probability only for **retrieving**, while refer to the **golden label** of retrieved anchors for making prediction.\
> Therefore, when test instance and its anchors are both wrongly predicted by ICL, $k$NN Prompting can still make correct prediction by referring to anchors' golden label. Such an exquisite design underpins the effectiveness of the proposed method.\
> We have elaborated and reorganized Section 4.3.3 to include this explanation.
>
> > The predictions are made based on majority voting of retrieved examples. This method **may suffer when the labels are imbalanced**. For example, when one label has much fewer examples compared to other labels, this label will not be favored by the majority voting system.
>
> This is a very insightful thought and probable hypothesis! We have conducted analysis accordingly.\
> We simulate the imbalance scenario on a binary classification task SST2, we sample one class proportionally to {0.5, 0.25, 0.125, 0.0625, 0.03125, 0.015625}, i.e., 0.5 corresponds to the balanced situation, and 0.015625 corresponds to extremely imbalanced situation, resulting in 32 negatives examples VS 2016 positives examples.\
> Indeed when one specific class have much more examples, $k$NN Prompting is overwhelmed by the large quantity of anchors, it is simply far more easier to find closer neighbors in one class than the other.\
> To address such performance degradation under challenging imbalanced scenario, we propose a **very simple normalization trick**: we average the anchor representation to produce **one** centered anchor for each class, the resulting anchor is thus more representative and also avoids quantity distraction.\
> Results show that such an adaptation works **surprisingly well** with **no loss of performance** even under ordinary balanced scenario.\
> We kindly refer the reviewer to Section 4.3.2 Figure 8 and Appendix C.4 Figure 13 for illustration. Results on SST2 are listed below:
>
> |Imbalance Ratio|0.015625|0.03125|0.0625|0.125|0.25|0.5|
> |--|--|--|--|--|--|--|
> |$k$NN Prompting|55.2|67.0|75.9|85.0|89.2|90.7|
> |$k$NN Prompting w/ Normalization|92.1|92.0|92.0|92.2|92.1|92.2|
>
> > Clarity & Quality: the paper is generally well-written and easy to follow. One thing confusing me is Table 1 -- I did not see authors mentioning Table 1 and the numbers in Table 1 seem contradictory to other tables.
>
> Thanks for the careful examination! The statistics are initially calculated with *average* instance length, which actually causes severe truncations. The statistics are referred to illustrate our motivation and have no impact on the experimental setting. They are now revised and can be found in Table 1 and Appendix A Table 6.
>
> |Method|SST2|SUBJ|MPQA|AGNews|CB|CR|DBPedia|MR|RTE|TREC|
> |--|--|--|--|--|--|--|--|--|--|--|
> |*Initially Reported*|
> |Maximum Num. of Shots (Calculated w/ Avg Instance Length)|53|29|98|8|7|35|2|31|12|19|
> |Truncation Prob|98%|93%|100%|100%|36%|96%|69%|96%|50%|100%|
> |*Revised Statistics*|
> |Maximum Num. of Shots for 2048 (Truncation Prob < 5%)|44|25|81|7|6|28|1|27|10|17|
> |$M_T$ for 2048 (Approximated into {2/4/8/16/32} for simplicity)|32|16|32|4|4|16|1|16|8|16|
> |Maximum Num. of Shots for 1024 (Truncation Prob < 5%)|20|12|39|3|2|14|1|14|4|8|
> |$M_T$ for 1024 (Approximated into {2/4/8/16/32} for simplicity)|16|8|32|2|2|8|1|8|4|8|

---

### Official Review · Reviewer_P4ZY · 2022-10-30

**Confidence:** 4
**Correctness:** 2
**Technical Novelty And Significance:** 2
**Empirical Novelty And Significance:** 2
**Recommendation:** 6

**Clarity, Quality, Novelty And Reproducibility:**

**Clarity**: Not good enough. The method is clearly introduced, but related works are not clearly mentioned or compared.

**Quality**: Not good enough.

**Novelty**: Low and not clearly compared with previous work.

**Reproducibility**: High.

**Strength And Weaknesses:**

# Strength
* This work proposes knn prompting that can help the large language models to better suit the application scenarios with enough training data rather than a few shots.
* For the proposed knn prompting, this work conducts further analysis and shows the robustness of this method. The qualitative analysis illustrates the explanations behind the performance of this method.

# Weakness
* **Lack of comparisons with previous retrieving-based in-context learning methods**. For the scenario considered in this work, a line of closely related work is about prompt retrieving, which also considers to store the extra training data in an additional memory and utilizes similarity-based choosing. I suppose the main difference between retrieving methods and knn prompting is that the former requires to store the whole data while the latter only need to store a probability distribution. But I suppose both of them do not go beyond your considered scenarios. For instance, [1] considers to retrieve prompting examples from the example bank with knn selection. Since you have mentioned this work in your related work, why not experimentally compare with it?
**In a word, a line of previous work can also deal with the challenge considered in this work, but this line is neither clearly mentioned nor experimentally compared**.

* **The basic idea behind knn prompting (i.e., calibration) is not clearly introduced and compared**. It seems that the the basic idea behind knn prompting is to calibrate the output probability of the model. Such an idea (along with in-context learning) has also been explored in many previous works, such as [2]. Actually, [2] is already mentioned in the paper but you didn't clearly show the relevance. **In a word, the basic idea of calibration should be clearly introduced to readers and further compared with previous works**.

* **Potential misleading on the novelty of this work**. It seems that this work wants to highlight its novelty on the considered application scenario (i.e., length limitation for in-context learning) and proposed methods (i.e., probability calibration for in-context learning).
But in my view, both two points are not novel enough (according to the two weakness mentioned above).
**For several highly related works, this work simply lists them but avoid to show a clear relevance and comparison, causing a potential misleading on its novelty**.

All in all, both the considered application scenario and the basic idea of proposed method in this work have been explored in many previous works, but neither the conceptual relevance nor the experimental comparison is mentioned.

[1] What makes good in-context examples for GPT-3?

[2] Calibrate before use: Improving few-shot performance of language models.

**Summary Of The Paper:**

This work considers application scenarios that utilize large language models in a in-context learning paradigm with training data that cannot fit the context due to the input-length limitation.
To alleviate the challenge posed by the maximum length limit of language models, this work proposes the knn prompting method.
The key idea of knn prompting is to store the information of a large amount of training data in an additional memory cache.
During the inference process, the output is generated by the nearest neighbor voting for a given input in the memory cache, rather than by the model directly.
This work claims that knn prompting can greatly improve in-context learning, especially in low-resource or fully
supervised scenarios.



**Summary Of The Review:**

There is a potential misleading on the novelty of this work, since neither the conceptual relevance to previous work nor the experimental comparison with previous work is clearly mentioned in this work.

---

> ### Author Response · Authors · 2022-11-19
> **Answers to Review P4ZY (1/2)**
>
> Thanks for such an insightful and constructive review. We have conducted according experiments as follows to make this a more comprehensive work and hopefully address the reviewer's concerns.
>
> > Lack of comparisons with previous retrieving-based in-context learning methods. ... For instance, [1] considers to retrieve prompting examples from the example bank with knn selection. ...
>
> ### Analytical Comparison
> Although both framed as *retrieval-based*, there exists an essential distinction between existing methods [1] and $k$NN Prompting, i.e., retrieve **before** LLM or **after** LLM.\
> Existing methods like [1] try to **select** demonstrations and **compose** the prompt (we thus refer them as **PromptCompose**), which potentially raises the following concerns:\
> **1).** The selection happens **before** LLM inference, and eventually still only in-context examples are exploited with LLM, while most of the training data are discarded in earlier stage, whose utility will be greatly undermined.\
> **2).** The reliance on an external retrieve model incurs further complications. Its implementation needs extra deliberation, its capability will be questioned and its meansurement for similarity might not suit the requirement of LLM prompting.\
> By contrast, $k$NN Prompting naturally leverages LLM itself to build datastore representations for retrieving and is free from the above weaknesses.
>
> ### Experimental Comparison
> We have reproduced according to [1] for comparison. The key findings are:\
> **1).** Although PromptCompose methods do exhibit marginal scaling benefits, $k$NN Prompting outperforms them with substantial improvements of [**+8.54**, **+10.19**] across all settings ({32, 64, 128, 256, 512, 1024}).\
> **2).** $k$NN Prompting performs on par with or even exceeds the approximate **upper bound** of PromptCompose.
> These conclusions are supported by the following results.
>
> * **$k$NN Prompting VS PromptCompose**\
> We employ various general-purpose sentence encoders including BM25, Sentence-BERT and state-of-the-art ones like SimCSE[2], Trans-Encoder[3].
> We first calculate the cosine similarity between test instance and all training data, then select top-k examples to compose the prompt.
> Results are excerpted as follows, more comprehensive results can be found in Section 4.2.3 Figure 6 and Appendix C.3 Table 8.
> |Method|SST2|SUBJ|MPQA|AGNews|CB|CR|DBPedia|MR|RTE|TREC|AVG|
> |--|--|--|--|--|--|--|--|--|--|--|--|
> |In-Context Learning|81.3|64.1|75.2|72.7|60.7|66.2|83.5|72.2|53.0|54.2|68.31|
> |$m=32$ (Training Data)|
> |BM25|68.4|63.6|75.2|69.0|65.0|55.2|80.7|59.4|53.0|65.5|65.50|
> |SBERT|71.0|67.5|75.2|82.3|62.9|57.8|83.8|57.7|51.2|58.4|66.78|
> |SimCSE|68.1|69.8|75.2|80.7|66.4|55.9|82.3|57.3|52.8|54.5|66.31|
> |Trans-Encoder|67.9|70.9|75.2|77.7|61.4|56.7|82.8|57.3|52.7|59.3|66.21|
> |$k$NN Prompting|87.7|77.0|75.2|86.2|58.9|88.2|94.1|83.9|53.6|64.8|**76.97 (+10.19)**|
> |$m=1024$ (Training Data)|
> |BM25|76.9|74.9|79.3|83.6|71.4|56.9|90.2|60.3|52.7|85.0|73.12|
> |SBERT|73.9|76.6|83.7|90.0|69.6|58.1|91.1|59.6|52.8|75.7|73.12|
> |SimCSE|73.5|74.5|84.6|88.7|73.2|57.5|91.6|59.8|53.1|69.4|72.59|
> |Trans-Encoder|73.4|78.4|82.9|87.8|73.2|55.9|91.7|59.5|54.8|76.6|73.42|
> |$k$NN Prompting|88.2|88.4|84.1|89.1|64.6|86.7|97.8|84.4|53.3|83.0|**81.96 (+8.54)**|
>
> * **$k$NN Prompting VS PromptCompose Upperbound**\
> We further push PromptCompose to an extreme situation trying to approximate its upper-bound.
> As PromptCompose methods ultimately resort to compose the prompt, their performance should be bounded by the best composition scheme from finite compositions.
> We search for **1,000** prompts with different examples and report the best run.
> These results is added in Section 4.2.3 Table 4.
> |Method|SST2|SUBJ|MPQA|MR|CR|TREC|
> |--|--|--|--|--|--|--|
> |In-Context Learning|81.3|64.1|75.2|72.2|66.2|54.2|
> |**Upper Bound** for Prompt Composition|**92.6**|86.0|**87.5**|**88.7**|**88.7**|73.0|
> |$k$NN Prompting|88.2|**88.4**|84.1|84.4|86.7|**83.0**|

---

> > ### Author Response · Authors · 2022-11-19
> > **Answers to Review P4ZY (2/2)**
> >
> > > The basic idea behind knn prompting (i.e., calibration) is not clearly introduced and compared. ...
> >
> > We have clarified the technical motivation of $k$NN Prompting in Abstract and Introduction, i.e., calibration-free optimization. We also added comparisons to state-of-the-art calibration-based methods [4][5] under strictly comparable few shot scenario, the results show that $k$NN Prompting generally outperforms calibration-based methods, which demonstrates the superiority of calibration-free optimization: it no longer needs to align general-domain LLM outputs with task-specific label space but exploit such outputs to align between test and anchor examples.\
> > The results are excerpted as follows, and we kindly refer the reviewer to comprehensive results in Section 4.2.1 Table 2 as well as the visualized scaling curve in Section 4.2.1 Figure 4 and Appendix C.1 Figure 12.
> > |Method|SST2|SUBJ|MPQA|MR|CR|TREC|AVG|
> > |--|--|--|--|--|--|--|--|
> > |$m=2$ (Training Data)|
> > |In-Context Learning|59.8|51.4|60.2|57.3|64.7|50.0|57.24|
> > |Contextual Calibration[4]|76.2|69.8|63.4|60.1|75.6|45.5|65.12|
> > |Noisy Channel[5]|82.6|64.6|60.2|83.3|79.4|35.2|**67.54**|
> > |$k$NN Prompting|77.5|73.4|56.6|69.7|81.1|41.3|66.57 (-0.97)|
> > |$m=4$ (Training Data)|
> > |In-Context Learning|67.2|56.5|70.8|60.8|62.8|50.9|61.50|
> > |Contextual Calibration|70.8|60.0|70.5|59.6|70.0|43.6|62.43|
> > |Noisy Channel|80.9|60.5|66.4|83.9|79.0|40.9|68.58|
> > |$k$NN Prompting|87.1|73.5|66.4|71.2|82.9|51.6|**72.14 (+3.29)**|
> > |$m=8$ (Training Data)|
> > |In-Context Learning|57.8|66.2|77.2|66.0|61.5|50.9|63.27|
> > |Contextual Calibration|68.5|64.5|72.7|64.9|68.2|44.0|63.80|
> > |Noisy Channel|82.0|62.5|70.1|85.0|79.2|41.6|70.06|
> > |$k$NN Prompting|88.9|77.7|72.5|75.4|84.6|63.7|**77.13 (+7.07)**|
> >
> > > Potential misleading on the novelty of this work. ... For several highly related works, this work simply lists them but avoid to show a clear relevance and comparison, causing a potential misleading on its novelty.
> >
> > We have rewritten the Abstract and Introduction to 1) more clearly present the novelty and contribution, and 2) include discussions of related works (calibration-based methods and PromptCompose methods).
> > The contribution is two-fold:
> > * In terms of application scenario novelty, $k$NN Prompting can effectively enable data scaling beyond the context with superior scaling factors, while existing PromptCompose methods are not even competitive.
> > * In terms of technique novelty, $k$NN Prompting is well motivated under the ICL paradigm to avoid forced alignment between LLM and label space, and is thus calibration-free.
> >
> > Both claims are supported and further strengthened by the updated experiments. We kindly refer the reviewer to the revised Abstract and Introduction section.
> >
> > [1] What Makes Good In-Context Examples for GPT-3? (Liu et al., DeeLIO 2022)\
> > [2] SimCSE: Simple Contrastive Learning of Sentence Embeddings (Gao et al., EMNLP 2021)\
> > [3] Trans-Encoder: Unsupervised sentence-pair modelling through self- and mutual-distillations (Liu et al., ICLR 2022)\
> > [4] Calibrate before use: Improving few-shot performance of language models. (Zhao et al., ICML 2021)\
> > [5] Noisy Channel Language Model Prompting for Few-Shot Text Classification. (Min et al., ACL 2022)

---

> > > ### Comment · Reviewer_P4ZY · 2022-11-26
> > > **Reply**
> > >
> > > Since the authors made huge changes in their revision (specifically new contributions and experiments), I'm sorry that I didn't have time to completely recheck the details of the new experimental setup and storytelling logic. Therefore, I simply assume that these additional parts are trustworthy.
> > >
> > > **For experiments**. I appreciate the additional comparisons! Since the authors claim that knn-prompting is superior to other sota methods (retrieval-based methods and calibration methods), I would like to increase my score.
> > >
> > > **For the claimed novelty**. A new point "calibration-free" is now highlighted, but I still do not consider this point as novel. Although the authors claim that knn-prompting is very different from calibration methods, but there is just an implementation (or technical) difference: voting for the final prediction (calibration-free), or directly changing the prediction distribution (calibration-based).
> > >
> > > **For the application**. I noticed another limitation in the application: knn-prompting cannot handle generative tasks (e.g., translations) in NLP. Therefore, the applicability of knn-prompting is greatly limited compared to other retrieval-based methods.
> > >
> > > All in all, I will update my score due to additional experimental results, but I still think this work has only a very limited contribution to the community due to the concerns about novelty and applicability.

---

> > > > ### Author Response · Authors · 2022-12-03
> > > > **Thank You for the Reply and More Follow-up Discussions**
> > > >
> > > > We are very thankful for the reviewer's appreciation of the additional experiments and the upgraded evaluation. And We would like to respectfully discuss on the follow-up concerns.
> > > >
> > > > ## On Novelty
> > > >
> > > > * We believe the calibration-free characteristic of $k$NN Prompting is one of our non-trivial contributions. Although both are motivated to resolve the misalignment between general-domain natural language and downstream task-specific label space, there exist essential differences between them:
> > > >     * **calibration-based** methods only try to **rectify or manipulate** the predicted probability distribution for categorical labels, they take various perspectives including probing the bias for each class label [1], permuting in-context demonstrations to reach balanced class label prediction [2], etc. This line of methods incurs several concerns:
> > > >         - **such manipulation can only alleviate the misalignment to a limited extent, because the LLM inevitably generates distributions over the entire vocabulary instead of label words only**;
> > > >         - they **rely on prior assumptions** that all target categories are balanced;
> > > >         - the rectification is made **based on merely part of the distribution** (i.e., manually defined label words), while the rest of the distribution (non-label words) is ignored, which are also implicitly informative.
> > > >     * Our **calibration-free** approach takes a much simpler but very different road which **avoids any manipulation**, and instead exploits the overall predicted probability distribution itself to perform inference. The key-value pair in the datastore describes the original, intrinsic correlation between LLM outputs distributions and target categories. Correspondingly:
> > > >         - **it no longer decodes generated distribution into label words, thus avoids input-label misalignment**;
> > > >         - it is **free from balanced assumption** of target categories (as validated in Figure 8);
> > > >         - it can **utilize the entire distribution** to precisely match test instances towards training data.
> > > >
> > > > * The claimed superiority is also well supported by our experimental results.
> > > >     * On strictly comparable few-shot setting, the proposed calibration-free method significantly outperforms existing calibration-based state-of-the-art results (Table 2 and Figure 4).
> > > >     * With increased training data, the proposed calibration-free method can effectively scale up, and continually optimize the collected label space distribution, producing further improved performance (Figure 5). For many practical scenarios with more than few-shot training data, the proposed method substantially outperforms calibration-based ones, as the latter can not benefit from beyond-context training data at all.
> > > >
> > > > We will add such discussions in a revised version to make this point more clear.
> > > >
> > > > ## On Applicability
> > > >
> > > > * In terms of task scope, we consider the contribution significant and meaningful, if not vastly extensive.
> > > >     * We verify the proposed method on 10 well-established datasets, which already covers a wide spectrum of downstream NLP application, including *Sentiment Analysis*, *Topic Classification*, and *Natural Language Inference*. These are common, typical and also very important scenarios for general natural language understanding.
> > > >     * The choice of benchmark is basically identical to many impactful previous works([2], [4], [3], inter alia). Thus this work shares similar contribution in terms of downstream task applicability.
> > > >     * On the investigated benchmark, the proposed method substantially outperforms existing SOTA results, and the PromptCompse (retrieval-to-compose) baselines are nowhere near competitive (Figure 6).
> > > >
> > > > * With all due respect, we believe the value of a work shall not be solely determined by the number of tasks it can address, but also its effectiveness on the situated scenario, its superiority compared to existing SOTA baselines, its simplicity that easily enables practical implementation, as well as its perspective of motivation that may further inspire future insights.
> > > >
> > > > * As a follow-up work and a valuable future direction, we will definitely explore the possibilities to further extend the proposed framework to more tasks.
> > > >
> > > > Finally, we would like to again express our appreciation for the insightful review and the pleasurable discussion.
> > > >
> > > > [1]Calibrate before use: Improving few-shot performance of language models. (Zhao et al., ICML 2021)\
> > > > [2]Fantastically Ordered Prompts and Where to Find Them: Overcoming Few-Shot Prompt Order Sensitivity (Lu et al., ACL 2022)\
> > > > [3]Noisy Channel Language Model Prompting for Few-Shot Text Classification. (Min et al., ACL 2022)\
> > > > [4]Making Pre-trained Language Models Better Few-shot Learners (Gao et al., ACL-IJCNLP 2021)

---

> > > > > ### Comment · Reviewer_P4ZY · 2022-12-06
> > > > > **Reply**
> > > > >
> > > > > I appreciate these further discussions! If the given detailed comparison between calibration-based and calibration-free methods could be more solidly analyzed, I would like to further increase my score.
> > > > >
> > > > > Here, I list some perspectives that authors could consider.
> > > > > - You mentioned that 'prior assumption' is important for calibration-based method but not for calibration-free method. In my view, the results of calibration-free method in Figure 8 also rely on the assumption that test set is balaenced.
> > > > > I suppose **there should be more detailed discussions, comparisons and analyses on the prior knowledge that two methods rely on.**
> > > > > I think there are mainly two kinds of prior knowledge: accessable distribution (i.e., the prior distribution for calibration-based method and the training set distribution for calibration-free method) and unaccessable distribution (i.e., test set distribution).
> > > > > **Does calibration-free still significantly outperform calibration-based when two methods share the same accessable distribution?**
> > > > > **How would calibration-free methods perform if test suit is also mismatched?**
> > > > > There are too many ways to make deeper explorations.
> > > > >
> > > > > - You mentioned that comparing with calibration-based method, the calibration-free method utilize the entire distribution which brings more information.
> > > > > If you could **concretely identify that the calibration-free method is truly benefit from more information in final predicted distribution**, this could make this paper impressive.
> > > > > For instance, **what would happen if the knn distance is only calculated on the label tokens rather than the whole predicted distribution?**
> > > > > This point is more important and insightful than the 'prior knowledge' one.
> > > > >
> > > > > Actually, for your revised paper, although you try to highlight the new contribution on 'calibration-free method', there is still limited insight if you could not further concretely reveal **why calibration-free method is better than calibration-based method** and **when calibration-free method works and when not**.
> > > > > If there is only a simple conclusion that 'calibration-free method has good performance', I could not buy in your contribution on exploring a novel calibration-free method due to the limited insight.

---

> > > > > > ### Author Response · Authors · 2022-12-09
> > > > > > **# Thank You for the Reply and Our Answers (1/3)**
> > > > > >
> > > > > > We are very grateful for such an insightful and constructive review! We try our best within the tight schedule to conduct the following comprehensive analyses accordingly.
> > > > > >
> > > > > > # 1. Reliance on Prior Knowledge
> > > > > > **We conduct comprehensive experiments as the reviewer instructed and achieve the following conclusions:**
> > > > > >  * The proposed calibration-free method **does not rely on or be affected by any prior knowledge regarding train / test distribution**.
> > > > > >  * Calibration-based method like ContextualCalibration[1] **indeed rely on the prior knowledge of test distribution**. While Calibration-based method like NoisyChannel[2] does not.
> > > > > >  * The proposed calibration-free method consistently provides state-of-the-art performance and **significantly outperforms calibration-based methods under imbalanced train, test or both**.
> > > > > >
> > > > > > We provide experimental settings, results and analyses as follows.
> > > > > >
> > > > > > ## 1.1 Experimental Settings
> > > > > > * Imbalanced Distribution
> > > > > >     * We investigate various combinations of imbalanced settings by controlling the imbalance ratio $\alpha$. For binary classification tasks, $\alpha_{train}/\alpha_{test}=0.125$ means one category accounts for 12.5% of the entire train / test set, and $\alpha=0.5$ corresponds to the balanced setting.
> > > > > >     * We investigate both $\alpha_{train}<0.5$ and $\alpha_{test}<0.5$, which results in four combinations of experimental settings in total, we report them in Table A, B, and C.
> > > > > > * Prior Knowledge of Imbalanced Distribution
> > > > > >     * **We investigate the prior knowledge it relies on specifically for calibration-based method ContextualCalibration,** as such methods necessarily assume a prior distribution to rectify the LLM predicted label word probabilities. Possible assumptions are:
> > > > > >         * *Balanced Prior Assumption*: simply assume a balanced distribution as prior knowledge, as designed in their original paper;
> > > > > >         * *Trainset Prior Assumption*: use the observed (accessible) train set distribution as prior knowledge.
> > > > > >         * Note that test distribution is unaccessible so we can not use it.
> > > > > >     * **Other methods (ICL, NoisyChannel and $k$NN Prompting) do not technically incorporate any prior knowledge.** So they are not concerned with this investigation dimension.
> > > > > > * For every setting, we investigate $5$ binary classification tasks (SST2, MPQA, SUBJ, MR, CR) and run with $10$ different random seeds, we report the average score of these $5*10$ results.
> > > > > > * We also report *MaxDrop* which measures the performance degradation compared to ordinary balanced setting ($\alpha=0.5$). We mark as *NoDrop* if there are no performance degradation.
> > > > > >
> > > > > > ## 1.2 Experimental Results
> > > > > > ### Table A. Imbalanced Train & Balanced Test ($\alpha_{train} < 0.5, \alpha_{test} = 0.5$)
> > > > > > |Imbalance Ratio ($\alpha_{train}$)|0.125|0.25|0.375|0.5 (Balanced)|AVG|MaxDrop|
> > > > > > |--|--|--|--|--|--|--|
> > > > > > |ICL|62.4|64.2|65.7|68.2|65.15|-5.8|
> > > > > > |ContextualCalibration w/ Balanced Prior Assumptinon|74.0|69.7|68.5|70.3|70.64|-1.8|
> > > > > > |ContextualCalibration w/ Trainset Prior Assumptinon|51.4|71.5|80.2|70.3|68.36|-18.9|
> > > > > > |NoisyChannel|70.0|70.9|71.9|72.6|71.33|-2.6|
> > > > > > |$k$NN Prompting|**79.0**|**80.1**|**80.5**|**80.3**|**79.98 (+8.65)**|**-1.3**|
> > > > > >
> > > > > > ### Table B. Balanced Train & Imbalanced Test ($\alpha_{train} = 0.5, \alpha_{test} < 0.5$)
> > > > > > |Imbalance Ratio ($\alpha_{test}$)|0.125|0.25|0.375|0.5 (Balanced)|AVG|MaxDrop|
> > > > > > |--|--|--|--|--|--|--|
> > > > > > |ICL|45.9|53.4|60.9|67.9|57.04|-22.0|
> > > > > > |ContextualCalibration w/ Balanced Prior Assumptinon|50.0|57.2|64.2|70.2|60.42|-20.2|
> > > > > > |ContextualCalibration w/ Trainset Prior Assumptinon|50.0|57.2|64.2|70.2|60.42|-20.2|
> > > > > > |NoisyChannel|72.1|72.2|71.8|72.4|72.14|**-0.6**|
> > > > > > |$k$NN Prompting|**77.1**|**78.0**|**78.7**|**79.7**|**78.38 (+6.24)**|-2.6|
> > > > > >
> > > > > > ### Table C. Imbalanced Train & Imbalanced Test ($\alpha_{train} = \alpha_{test} < 0.5$)
> > > > > > |Imbalance Ratio ($\alpha_{train}=\alpha_{test}$)|0.125|0.25|0.375|0.5 (Balanced)|AVG|MaxDrop|
> > > > > > |--|--|--|--|--|--|--|
> > > > > > |ICL|45.8|50.0|58.6|67.9|55.57|-22.1|
> > > > > > |ContextualCalibration w/ Balanced Prior Assumptinon|63.0|57.6|61.7|70.2|63.13|-12.6|
> > > > > > |ContextualCalibration w/ Trainset Prior Assumptinon|**87.8**|**84.5**|78.8|70.2|**80.33**|**NoDrop**|
> > > > > > |NoisyChannel|64.6|68.6|70.8|72.4|69.10|-7.8|
> > > > > > |$k$NN Prompting|79.8|80.7|**80.3**|**79.7**|80.12 (-0.21)|**NoDrop**|
> > > > > >
> > > > > > ### Balanced Train & Balanced Test ($\alpha_{train} = \alpha_{test} = 0.5$)
> > > > > > This falls back to ordinary setting, and is already reported and concluded in the paper.

---

> > > > > > > ### Author Response · Authors · 2022-12-09
> > > > > > > **# Thank You for the Reply and Our Answers (2/3)**
> > > > > > >
> > > > > > > ## 1.3 Experimental Analysis
> > > > > > >  * For the proposed calibration-free method:
> > > > > > >      * **Technically, it does not incorporate any prior knowledge of train or test distribution**. Both the construction of datastore and the retrieving and prediction procedure **do not change** w.r.t. different prior knowledge of distribution.
> > > > > > >      * Without any prior knowledge and assumption, the proposed method can consistently provide state-of-the-art performance under all imbalanced settings. **Compared to other methods, the improvements are significant** (**+8.65/+6.24/-0.21** respectively in Table A, B, C).
> > > > > > >      * Without any prior knowledge and assumption, the proposed method can robustly adapt to all imbalanced settings, including imbalanced trainset, testset and both. **Compared to its own performance under balanced setting, there is basically no performance degradation** (**-1.3/-2.6/NoDrop** respectively in Table A, B, C, where -1.3/-2.6 can be recognized within ordinary fluctuation).
> > > > > > >  * For calibration-based methods like ContextualCalibration:
> > > > > > >      * Technically, it necessarily requires prior knowledge to rectify the LLM predicted label word probabilities. **If the prior knowledge does not match the (unaccessible) test distribution, its performance will be greatly degraded**.
> > > > > > >      * If *Balanced Prior Assumption* is consistent with test distribution, the method performs well (Table A, **-1.8**), otherwise, the performance degrades (Table B, **-20.2** and Table C, **-12.6**).
> > > > > > >      * Similarly, if *Trainset Prior Assumption* is consistent with test distribution, the method performs well (Table C, **NoDrop**), otherwise, the performance degrades (Table B, **-20.2** and Table A, **-18.9**)
> > > > > > >  * For calibration-based methods like NoisyChannel:
> > > > > > >      * Technically, it also does not incorporate any prior knowledge. It reformulates the input-label prompting as label-input prompting to calibrate the predicted probability.
> > > > > > >      * It generally performs well under imbalanced settings (-2.6/-0.6/-7.8). Nonetheless, its performance are still significantly inferior compared to the proposed method across all settings.

---

> > > > > > > > ### Author Response · Authors · 2022-12-09
> > > > > > > > **# Thank You for the Reply and Our Answers (3/3)**
> > > > > > > >
> > > > > > > > # 2. Benefits of Whole Predicted Distribution Utilization
> > > > > > > >
> > > > > > > > ## 2.1 Analyses
> > > > > > > > **We concretely identify whole predicted distribution as one key aspect of the benefits of the proposed method**, which is much more informative and represents the overall understanding of LLM for the test instance.\
> > > > > > > > By contrast, calibration-based methods as well as standard ICL only access the distribution over **label words**, which is incomplete in two aspects:
> > > > > > > >  * **Loss of information**. LLM considers all possible words in the entire vocabulary, non-label words probabilities also reflect its **understanding** for the test instance in certain perspectives, and thus are very informative.
> > > > > > > >  * **Multiple label words competing with each other**. There exist various choices for label words but no oracle rules to select one, and alternative choices potentially compete with the selected label words, distorting the label space distribution. This is also referred to as *surface form competition*, as discussed in a previous work [3].
> > > > > > > >
> > > > > > > > ## 2.2 Experiments
> > > > > > > > We mask out the non-label words probability, while leaving the rest of the method unchanged. We achieve the following conclusions:
> > > > > > > >  * When considering the whole predicted distribution, the proposed method exhibits crucial benefits, and significantly outperforms label-only distribution or calibration-based methods (**-0.97/+3.56/+7.07/+6.36/+8.50**).
> > > > > > > >  * When considering label-only distribution, the proposed method is only competitive or slightly better than calibration-based methods (-2.81/+2.42/+2.86/+1.23/+1.27).
> > > > > > > >
> > > > > > > > |Method|SST2|SUBJ|MPQA|MR|CR|TREC|AVG|
> > > > > > > > |--|--|--|--|--|--|--|--|
> > > > > > > > |$m=2$ (Training Data)|
> > > > > > > > |In-Context Learning|59.8|51.4|60.2|57.3|64.7|50.0|57.24|
> > > > > > > > |Contextual Calibration|76.2|69.8|63.4|60.1|75.6|45.5|65.12|
> > > > > > > > |Noisy Channel|82.6|64.6|60.2|83.3|79.4|35.2|**67.54**|
> > > > > > > > |$k$NN Prompting **w/ Whole Distribution**|77.5|73.4|56.6|69.7|81.1|41.3|66.57(-0.97)|
> > > > > > > > |$k$NN Prompting **w/ Label-Only Distribution**|77.8|68.9|53.3|66.7|81.6|40.2|64.73(-2.81)|
> > > > > > > > |$m=4$ (Training Data)|
> > > > > > > > |In-Context Learning|67.2|56.5|70.8|60.8|62.8|50.9|61.50|
> > > > > > > > |Contextual Calibration|70.8|60.0|70.5|59.6|70.0|43.6|62.43|
> > > > > > > > |Noisy Channel|80.9|60.5|66.4|83.9|79.0|40.9|68.58|
> > > > > > > > |$k$NN Prompting **w/ Whole Distribution**|87.1|73.5|66.4|71.2|82.9|51.6|**72.14(+3.56)**|
> > > > > > > > |$k$NN Prompting **w/ Label-Only Distribution**|85.9|70.5|67.4|68.1|82.0|52.2|71.00(+2.42)|
> > > > > > > > |$m=8$ (Training Data)|
> > > > > > > > |In-Context Learning|57.8|66.2|77.2|66.0|61.5|50.9|63.27|
> > > > > > > > |Contextual Calibration|68.5|64.5|72.7|64.9|68.2|44.0|63.80|
> > > > > > > > |Noisy Channel|82.0|62.5|70.1|85.0|79.2|41.6|70.06|
> > > > > > > > |$k$NN Prompting **w/ Whole Distribution**|88.9|77.7|72.5|75.4|84.6|63.7|**77.13(+7.07)**|
> > > > > > > > |$k$NN Prompting **w/ Label-Only Distribution**|88.9|69.2|67.8|72.8|84.3|54.6|72.92(+2.86)|
> > > > > > > > |$m=16$ (Training Data)|
> > > > > > > > |In-Context Learning|67.7|75.5|77.6|73.3|61.8|52.0|67.97|
> > > > > > > > |Contextual Calibration|75.7|59.7|75.2|73.0|73.1|46.5|67.21|
> > > > > > > > |Noisy Channel|84.4|62.4|70.4|83.7|79.6|54.2|72.44|
> > > > > > > > |$k$NN Prompting **w/ Whole Distribution**|88.8|80.9|68.2|80.1|84.8|70.0|**78.80(+6.36)**|
> > > > > > > > |$k$NN Prompting **w/ Label-Only Distribution**|89.7|71.4|60.5|79.8|84.8|55.8|73.67(+1.23)|
> > > > > > > > |$m=32$ (Training Data)|
> > > > > > > > |In-Context Learning|66.5|70.1|77.4|67.7|63.6|52.0|66.22|
> > > > > > > > |Contextual Calibration|76.9|58.6|76.5|78.5|71.2|44.3|67.66|
> > > > > > > > |Noisy Channel|84.8|61.1|70.8|82.5|80.0|47.6|71.14|
> > > > > > > > |$k$NN Prompting **w/ Whole Distribution**|89.0|83.2|69.3|77.8|85.0|73.5|**79.64(+8.50)**|
> > > > > > > > |$k$NN Prompting **w/ Label-Only Distribution**|86.7|65.9|64.4|74.2|83.9|59.4|72.41(+1.27)|
> > > > > > > >
> > > > > > > > The above conclusion invariantly holds for fully supervised setting.
> > > > > > > > |Method|SST2|SUBJ|MPQA|AGNews|CB|CR|DBPedia|MR|RTE|TREC|AVG|
> > > > > > > > |-|-|-|-|-|-|-|-|-|-|-|-|
> > > > > > > > |ICL ($m=M_T$)|81.3|64.1|75.2|72.7|60.7|66.2|83.5|72.2|53.0|54.2|68.31|
> > > > > > > > |$k$NN Prompting **w/ Whole Distribution** ($m=1024$)|88.2|88.4|84.1|89.1|64.6|86.7|97.8|84.4|53.3|83.0|**81.96(+13.65)**|
> > > > > > > > |$k$NN Prompting **w/ Label-Only Distribution** ($m=1024$)|87.7|69.3|84.5|63.2|84.1|96.8|83.4|84.7|49.0|65.7|76.84(+8.53)|
> > > > > > > >
> > > > > > > > # 3. MISC
> > > > > > > >  * We will add all the above results and discussions in a revised version appropriately.
> > > > > > > >  * Again, we are very thankful for the reviewer's expertise, valuable time, insightful feedback and active engagement throughout the discussion stage, which have greatly improved the thoroughness of this paper.
> > > > > > > >  * As the discussion stage is now ending soon, we look forward to the reply and would like to see if there are any further concerns we can try to timely address.
> > > > > > > >
> > > > > > > >
> > > > > > > > [1] Calibrate before use: Improving few-shot performance of language models. (Zhao et al., ICML 2021)\
> > > > > > > > [2] Noisy Channel Language Model Prompting for Few-Shot Text Classification. (Min et al., ACL 2022)\
> > > > > > > > [3] Surface Form Competition: Why the Highest Probability Answer Isn’t Always Right (Holtzman et al., EMNLP 2021)

---

> > > > > > > > > ### Comment · Reviewer_P4ZY · 2022-12-11
> > > > > > > > > **Reply**
> > > > > > > > >
> > > > > > > > > I increase my socre becasue these results and analysis could provide more insights behind knn-prompting, thus alleviating my concerns about the novelty of the paper.
> > > > > > > > > At last, I encourage authors to fully refine their paper as they promised.

---

> > > > > > > > > > ### Author Response · Authors · 2022-12-13
> > > > > > > > > > **Thank You for the Timely Reply!**
> > > > > > > > > >
> > > > > > > > > > We thank the reviewer for the timely reply, and all the valuable time he / she has spent throughout the discussion stage to greatly improve the paper.
> > > > > > > > > >
> > > > > > > > > > We will certainly include all the above contents (discussions / experimental results) and carefully organize them in a revised version!

---

### Author Response · Authors · 2022-11-19
**General Response on Updates**

We thank all reviewers for their insightful and constructive comments, we have made the following revisions to present a more comprehensive work:
* We have rewritten the **Abstract** and **Introduction** to strengthen and clarify the contribution of this paper, which is 1) superior calibration-free optimization and 2) effective beyond-context data scaling capability, discussions on related works are also added accordingly.
* We reproduced PromptCompose for comparison (**Figure 6**), such methods resort to compose prompt via example selection.
* We reproduced state-of-the-art calibration-based methods for comparison under comparable few shot scenario (**Table 2 & Figure 4**).
* We challenge the proposed method with extremely imbalanced data setting (**Figure 8**).
* We expanded and organized the explanation of effectiveness to take a further step to understand its intrinsic mechanism (**Section 4.3.3**).

Besides, to make room for the above experiments, some of the ablations (on distance measurement and others.) are now adjusted to **Appendix B**.\
All code to reproduce the above experiments will also be included in the final opensourced repo.

---

### Decision · Program_Chairs · 2023-01-20

**Decision:**

Accept: poster

**Justification For Why Not Higher Score:**

The limitations in terms of novelty make it less appealing as a spotlight/oral

**Justification For Why Not Lower Score:**

Despite the novelty limitations, the paper shows meaningful insights that will be of interest to the community

**Metareview: Summary, Strengths And Weaknesses:**

The paper addresses the problem of solving classification problems for NLP tasks by via prompting LLMs. The technique uses the LLM to map an example to a space in which a simple KNN solver works well. The paper is very well written, and its very easy to understand both the methods and the experiments. The experiments are very thdorogouh and are convincing in that the method is superior to the baselines, and provides an easy to use (and tune) method to take advantage of LLMs both for few-shot and larger training sets.

An issue that caused some disagreement among the reviews is that of novelty. Previous papers have already explored the idea of using a language model to map an example into a vector space, then apply KNN for classification. The distinction is the setting where the method is applied, the experiments performed, and baselines. The extensive experiments explaining exactly how this method can be applied and map it in the setting of LLMs seems to be new in this work. Overall, despite the somewhat limited novelty in terms of the method itself, I found (and this is supported by most reviews) that the study showing how it can be applied will be interesting to the research community. Considering this and the other advantages of the paper, I believe it will be a welcome addition to ICLR.

**Note From Pc:**

if the above contains the word "oral" or "spotlight" please see: "oral" presentation means -> notable-top-5% and "spotlight" means -> notable-top-25%. As stated in our emails, we are disassociating presentation type from AC recommendations